# Graph-Supported Dynamic Algorithm Configuration for Multi-Objective Combinatorial Optimization

## Abstract

Deep reinforcement learning (DRL) has been widely used for dynamic algorithm configuration, especially for evolutionary algorithms, which benefit from adaptive update of parameters during the algorithmic execution. However, applying DRL to algorithm configuration for multi-objective combinatorial optimization (MOCO) problems remains relatively unexplored. This paper presents a novel graph neural network (GNN) based DRL to configure multi-objective evolutionary algorithms. We model the dynamic algorithm configuration as a Markov decision process, representing the convergence of solutions in the objective space by a graph, with their embeddings learned by a GNN to enhance the state representation. Experiments on diverse MOCO challenges indicate that our method outperforms traditional and DRL-based algorithm configuration methods in terms of efficacy and adaptability. It also exhibits advantageous generalizability across objective types and problem sizes, and prospective applicability to different evolutionary algorithms.

## 1 Introduction

Selecting the right hyperparameters is crucial for the performance of optimization algorithms. Some automated algorithm configuration (AC) methods (López-Ibáñez et al., 2016; Lindauer et al., 2022) have been developed to identify well-performing configurations and reduce the need for labor-intensive trial-and-error tuning. As the optimal parameter values may change throughout different stages of the algorithmic deployment (Aleti, 2012), various dynamic algorithm configuration (DAC) methods have been proposed in recent years (Biedenkapp et al., 2020; Adriaensen et al., 2022). DAC adjusts the configuration of algorithms in real time, which is advantageous for algorithms facing changes in the search space configuration during execution. This adaptability to update parameters is especially relevant to iterative algorithms, such as Evolutionary Algorithms (EAs), a prominent class of Evolutionary Computation (EC) techniques for solving complex optimization problems. The performance of EAs relies significantly on the precise adjustment of their parameters and may require changes at various phases of the search process to maintain optimal performance.

Deep Reinforcement Learning (DRL) has been successfully used to control parameter values for various single-objective EC algorithms in different domains, as reported in the literature (Sharma et al., 2019; Sun et al., 2021; Tan & Li, 2021). These approaches address the parameter configuration problem by modeling it as a contextual Markov decision process (MDP) (Biedenkapp et al., 2020). This enables dynamic algorithm configuration to be approached as a sequential decision-making problem, enabling Reinforcement Learning (RL) to control algorithm configurations during search. Xue et al. (2022) extend the existing DRL-based DAC approaches to tackle multi-objective optimization. Although these methods have demonstrated their effectiveness in configuring parameters during the search, their applications are primarily limited to (multi-objective) continuous optimization, such as tuning hyperparameters of machine learning models, as in AutoML (Biedenkapp et al., 2020; Eimer et al., 2021), and benchmarking continuous functions (Xue et al., 2022).

In this paper, we propose a DRL-based, dynamic algorithm configuration method designed specifically for solving multi-objective combinatorial optimization problems (MOCOs). Most (multi-objective) combinatorial optimization problems, such as machine scheduling, vehicle routing, and resource allocation problems, are NP-hard, as they involve finding high-quality solutions in a large

space of discrete decision variables. Hence, practical approaches for solving these problems typically rely on heuristics, among which EAs have been widely used in various COPs (Bartz-Beielstein et al., 2014; Zhou et al., 2011).

We expect (and confirm with experiments in Section 4.1) that the existing DAC approach designed for continuous optimization (i.e., MADAC (Xue et al., 2022)) may not work well on large-size, complex COPs with many objectives, due to less smooth solution spaces and a wide range of objective values of COPs. To tackle these challenges, our proposed method, called GS-MODAC, employs a Graph Neural Network to capture the state of the search algorithm. Specifically, we take inspiration from various convergence- and diversity–based metrics for multi-objective optimization, such as the number of elite solutions, the spacing between solutions, the relative size of holes (gaps) in the solution space, and hypervolume. With this, we expect that our method leverages the graph-based representation to dynamically learn similar (yet advanced) features during the optimization process to reflect the current state in the multiple objective planes. By representing the state space as a graph, our method provides a state configuration independent of the number of objectives, eliminating the need for practitioners to configure arbitrary state features manually. In addition, GS-MODAC leverages a rewarding scheme designed to be incentivized toward Pareto optimal solutions in a problem-agnostic manner, fostering generalizability between differently scaled COPs.

Experimentation demonstrates that GS-MODAC is better than state-of-art algorithm configuration methods based on heuristics (irace) (López-Ibáñez et al., 2016), Bayesian Optimization (SMAC3) (Lindauer et al., 2022), and a multi-agent DRL approach (MADAC) (Xue et al., 2022). We further demonstrate that the proposed method can be applied to multiple Multi-Objective Evolutionary Computation algorithms to solve different MOCOs from distinct problem domains featuring varying numbers of objectives. Also, the trained models can generalize to effectively solve instances of larger sizes and more constrained problem variants, which were not observed in training.

Our study offers the following contributions:

1) We introduce GS-MODAC, a GNN and DRL-based method to dynamically control the parameter configuration of MOEAs for solving MOCOs. This approach effectively addresses the limitations of static algorithm configuration methods, achieving better convergence and more diverse solutions.

2) We propose a graph representation of solutions in the objective space, which is learned by graph neural network and involved in the state. Based on the normalized objectives, we also present an instance-agnostic reward function that applies to problems of different types and varying sizes.

3) We evaluate the proposed method on typical routing and scheduling problems and demonstrate its promising generalizability to perform effectively on more constrained problem variants and larger problem instances not encountered during training.

## 2 BACKGROUND AND RELATED WORK

**Multi-Objective Optimization (MOO) and Combinatorial Optimization (MOCO).** Combinatorial Optimization is concerned with finding the best solution from a finite set of feasible solutions. These problems are characterized by their discrete nature, where the solutions can be represented as integers, graphs, sets, or sequences. Multi-Objective Combinatorial Optimization (MOCO) involves simultaneously optimizing multiple, often conflicting objectives for combinatorial optimization problems. The general formulation of MOCO can be expressed as $\min_{x \in X} f(x) = (f_1(x), f_2(x), \ldots, f_N(x))$. Here, $X$ denotes the set of feasible solutions, $N$ is the number of objective functions to be optimized, and each $f_i(x)$ represents an objective function to be minimized.

**Definition 1: Pareto Dominance.** A solution $x_1 \in X$ dominates another solution $x_2 \in X$ ($x_1 \prec x_2$) if and only if: $f_i(x_1) \leq f_i(x_2)$ for all $i \in \{1, \ldots, N\}$, and there exists at least one $j \in \{1, \ldots, N\}$ such that $f_j(x_1) < f_j(x_2)$.

**Definition 2: Pareto Optimality.** A solution $x^* \in X$ is considered Pareto optimal if there is no other solution $x' \in X$ satisfying $x' \prec x^*$. In other words, $x^*$ is Pareto optimal if it is not dominated by any other solution in $X$.

**Definition 3: Pareto Front.** The objective of multi-objective optimization is to find the Pareto front, which consists of all Pareto-optimal solutions: $\mathcal{P} = \{x^* \in X \mid \nexists x' \in X \text{ such that } x^* \prec x'\}$. The

corresponding Pareto front is defined as: $\mathcal{F} = \{f(x) \mid x \in \mathcal{P}\}$. The Pareto front consists of the objective values of the Pareto set, where each $f(x)$ represents a point in the objective space.

**Definition 4: Hypervolume Indicator.** The Hypervolume (HV) indicator is a widely used metric for assessing performance in multi-objective optimization problems, providing a comprehensive evaluation of both convergence and diversity, even without knowledge of the exact Pareto front (Zitzler & Thiele, 1998). For a Pareto front $\mathcal{F}$ in the objective space, the HV with respect to a fixed reference point $r \in \mathcal{R}^N$ is defined as:

$$\text{HV}_r(\mathcal{F}) = \mu \left( \bigcup_{f(x) \in \mathcal{F}} [f(x), r] \right) \tag{1}$$

where $\mu$ denotes the Lebesgue measure, representing the $N$-dimensional volume, and $[f(x), r]$ refers to an $N$-dimensional cube: $[f(x), r] = [f_1(x), r_1] \times [f_N(x), r_N]$, which spans the region in the $N$-dimensional objective space between a point on the Pareto front and a fixed reference point $r$.

**Algorithm Configuration.** Algorithm Configuration (AC) involves determining optimal parameter configurations for an algorithm to maximize performance across various inputs. Dynamic Algorithm Configuration (DAC) extends AC by adjusting parameters during the optimization process to enhance performance (Biedenkapp et al., 2020). Unlike static configurations, DAC aims to balance exploration and exploitation, increasing the likelihood of finding high-quality solutions. According to Karafotias et al. (2014), it can be classified into three types: 1) *Deterministic*, which changes parameter configurations based on a predetermined rule, often using a time-varying schedule (Sun et al., 2020); 2) *Self-adaptive*, integrating parameter adjustments into the search process, allowing parameters to evolve alongside solutions (Michalewicz et al., 2000); and 3) *Adaptive parameter control*, which adjusts parameters based on search feedback, using credit assignment and operator selection to optimize performance (Aleti & Moser, 2016).

Machine Learning methods such as Bayesian Optimization and Artificial Neural Networks have been used to tune parameters by predicting parameter performances based on training instances ( Lessmann et al. (2011); Biswas et al. (2021); Centeno-Telleria et al. (2021)). In recent years, there has been a significant focus on using Reinforcement Learning (RL) for dynamic algorithm configuration, especially for controlling parameters in evolutionary algorithms (EAs). RL enables the learning of dynamic policies that adapt to the evolving state of the problem. In contrast to traditional optimization methods, which typically rely on fixed parameters and lack the ability to adjust based on the context, RL allows agents to continuously interact with the environment and update their decision-making strategy as new information becomes available. This ability to adjust over time makes it particularly well-suited for problems where the optimal solution evolves or depends on changing conditions (Biedenkapp et al., 2020). In ECs, different parameter configurations can be treated as a set of actions, and when a configuration set leads to improved solutions, a reward is given to the RL agent. Recent research has demonstrated the effectiveness of RL in controlling the parameters of EAs. For example, Q-learning has been applied to adapt each generation's crossover and mutation rates to solve a vehicle routing problem (Quevedo et al., 2021). Similarly, an EA has been hybridized with state–action–reward–state–action (SARSA) and Q-Learning to control crossover and mutation rates for the Flexible Job Shop Scheduling Problem (Chen et al., 2020). There has also been an increasing interest in using Deep Reinforcement Learning (DRL). Examples are the employment of a Double Deep Q-Network (DDQN) agent to select parameters in Differential Evolution (DE) (Sharma et al., 2019) or a Policy Gradient method (Sun et al., 2021). Several works have extended RL-based DAC methods to address multi-objective optimization (Huang et al., 2020; Tian et al., 2022; Reijnen et al., 2022; Han et al., 2023). However, applying DRL in multi-objective optimization presents several challenges. Many existing approaches rely on manually configured features derived from convergence and fitness landscapes, such as the number of elite solutions, solution spacing, the relative size of gaps in the solution space, and hypervolume, to define the states in the MDP. This process is labor-intensive and often suboptimal. Additionally, managing high-dimensional configured state spaces and optimizing for multi-objectives complicates the learning process (Yang et al.). Moreover, most studies focus on search operator selection, typically configured as discretized actions, and are often trained and demonstrated on simple continuous optimization problems and standard benchmark functions (Ma et al., 2024).

The closest work to ours is Xue et al. (2022), where the authors propose MADAC for tuning parameters in a multi-objective evolutionary algorithm (MOEA). The work utilizes value-decomposition

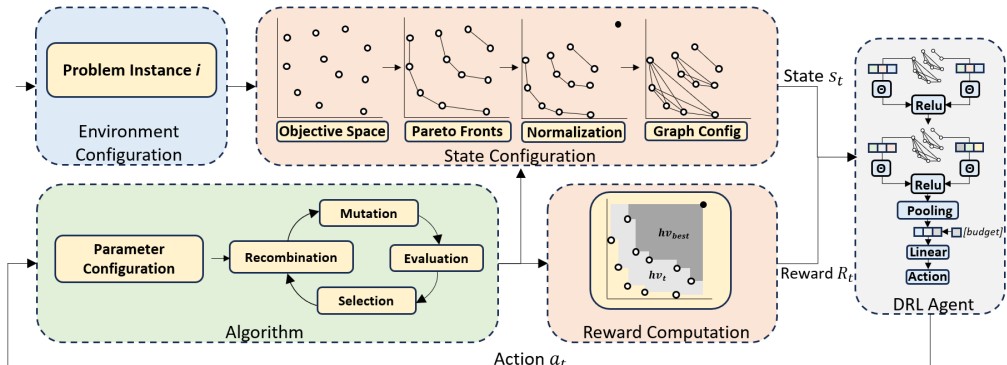

Figure 1: The GS-MODAC framework. The DRL agent chooses actions to configure the next iteration of the algorithmic search based on the learned graph embedding of the state. Nodes represent the normalized objectives of solutions at a given iteration of the search on multiple objective planes. The graph is constructed by interconnecting the normalized objective points in the different Pareto fronts, creating a structured visualization of the solution space. The actions are performed in the environment, which in response returns the next generation of solutions, the next state, and the reward.

networks (VDN) (Sunehag et al., 2017), a typical multi-agent RL method, to identify the optimal settings for different categories of parameters. The work incorporates information from the specific problem instance, the ongoing optimization process, and the evolving population of solutions. The reward function incentivizes improvement, offering rewards for discovering better solutions and greater rewards for further advancements in later stages. The limitation of MADAC is that it typically includes information on convergences, objectives, and population-based metrics based on arbitrary hand-defined and tuned state features. This reliance on manually selected features can lead to suboptimal results, as the chosen features may not adequately capture the complexity of the environment. To address this, we propose a novel DRL-based approach for the dynamic configuration of parameters in MOEAs aimed at solving Multi-objective Combinatorial Optimization problems. Instead of relying on arbitrarily defined features, our approach involves mapping the objective spaces to graph structures and utilizing Graph Neural Networks (GNNs) to aggregate node features as states. This method allows for a more comprehensive and adaptive representation of the state space, scalable to multiple objective problems, potentially enhancing the performance and robustness of the evolutionary algorithms.

## 3 THE METHOD

This section presents our proposed method, GS-MODAC (Graph-Supported Multi-Objective Dynamic Algorithm Configuration). GS-MODAC employs a Graph Neural Network (GNN) to capture the state of the search algorithm and Deep Reinforcement Learning (DRL) to configure the next search iteration in solving MOCOs. Graphs offer a powerful means of representing structured and informative embeddings, with the flexibility to scale to different sizes. GNNs, in particular, have demonstrated significant versatility and effectiveness across diverse graph-related tasks due to their ability to model complex graph structures and extract meaningful representations (Zhou et al., 2020). In this work, we leverage GNNs to extract the graph state, enabling the DRL agent to make more informed and effective decisions based on the iterative search's current state. By representing the state space as a graph, our method utilizes a state configuration independent of the number of objectives, bypassing the need for practitioners to customize state representations for any number of objectives manually. We illustrate the overview of GS-MODAC in Figure 1.

### 3.1 MDP FORMULATION FOR GS-MODAC

GS-MODAC is built upon the foundation of Dynamic Algorithm Configuration (DAC) principles (Biedenkapp et al., 2020), dynamically adjusting parameter configuration of EAs during their optimization processes. This process can be formulated as a contextual Markov Decision Process

(MDP) $M_I$, with shared action and state spaces, but with different transition and reward functions for each instance $i$ in a set $I$. Each $M_i$ corresponds to the MDP of a specific problem instance $i$, encapsulating the state space $S$, action space $A$, state transition function $T_i$, and reward function $R_i$.

In the context of GS-MODAC, given a target algorithm with the space of its configuration hyperparameters $\Theta$, a policy $\pi$ maps the state $s \in S$ to action $a \in A$ (i.e., hyperparameter configuration $\theta \in \Theta$). The primary objective is training the policy to enhance the algorithmic performance across a diverse set of instances, minimizing the expected cost function $c(\pi, i)$ across instances $i \in I$. To further facilitate generalizability, we define a shared reward function $R$ to consistently measure performance improvement across different problem instances. This shared reward function $R$ ensures that the policy learns to optimize the performance of the target algorithm to make it generalize well across various instances rather than overfitting to specific instances. We introduce the components composed within the MDP of GS-MODAC as follows.

**States.** The state space $S$ provides a DRL agent with information on the current status of the search algorithm, aiding in selecting the best action for the next iteration. In the context of DAC, several studies have attempted to create a state configuration that accurately represents the convergence process and generalizes to unexplored problem instances (Sharma et al., 2019; Sun et al., 2020; Xue et al., 2022). These configurations typically include convergence information, objective values, and population diversity metrics. In contrast to the literature, we innovatively propose mapping objective spaces to graphs and leveraging GNNs to dynamically learn state representations. The graph transformation of objective space is illustrated in 'state configuration' in Figure 1.

This transformation involves interconnecting normalized objective points in the different Pareto fronts to create a structured visualization of the solution space and eliminates the need for manual state space design, a process known to be cumbersome and suboptimal. To ensure the state configuration is independent of the magnitude of objective values, we normalize the solution space relative to a reference point that is defined by the worst observed objective values in the first population of solutions. In doing so, we provide a state configuration that effectively represents the algorithmic convergence and the diversity of solution performances, potentially generalizing to problem instances with varying objective magnitudes. An additional feature vector is correspondingly included, containing the normalized number of generations that have been passed by, representing the remaining budget available for the search.

**Actions.** The action space $A$ consists of multiple continuous values, each associated with an evolutionary algorithm parameter to be controlled. These values are normalized between -1 and 1, defined based on the recommended values from rules of thumb for EA tuning (Coello et al., 2007).

**Transitions.** The transition function outlines the dynamics of the search algorithm and is led by interactions between the agent and the problem environment. In the context of GS-MODAC, each interaction (step) with the environment serves as a search iteration. Given state $s_t$, an agent takes a action $a_t$, and the probability of moving to state $s_{t+1}$ is denoted as $T(s_{t+1}|s_t, a_t)$. Unlike the state, action, and reward spaces (in the scope of this work), the transition function is contingent upon the specific instance $i \in I$.

**Rewards.** The reward function is critical in guiding policy learning. In multi-objective optimization, the rewarding system should encourage algorithmic convergence towards the optimal Pareto front. However, the evolving towards the Pareto front often turns increasingly demanding along search steps. The early search stages typically allow for swift gains, while the later stages require substantially more effort. In light of this, we design rewards for enhancing the evolvement of the Pareto front in the latter.

In specific, we design the reward function as follows: At each iteration $t$, we assess whether the hypervolume of the population $HV_{\text{current}}$ exceeds the best previously observed hypervolume $HV_{\text{best}}$. If $HV_{\text{current}} > HV_{\text{best}}$, we compute the percentage improvements, i.e., $\Delta_{\text{current}}$ and $\Delta_{\text{best}}$, and then calculate the reward as the difference between the squared improvements. In this way, we magnify the rewards for larger improvements in later stages, encouraging significant evolvement of the Pareto front. The reward is defined as:

$$r_t = \begin{cases} \Delta_{\text{current}}^2 - \Delta_{\text{best}}^2 & \text{if } HV_{\text{current}} > HV_{\text{best}} \\ 0 & \text{otherwise} \end{cases}$$

where $\Delta_{\text{current}}$ and $\Delta_{\text{best}}$ are calculated as follows:

$$\Delta_{\text{current}} = \left( \frac{HV_{\text{current}} - HV_{\text{initial}}}{HV_{\text{ideal}} - HV_{\text{initial}}} \right) \times 100, \quad \Delta_{\text{best}} = \left( \frac{HV_{\text{best}} - HV_{\text{initial}}}{HV_{\text{ideal}} - HV_{\text{initial}}} \right) \times 100$$

Hypervolumes are calculated using a nadir point, defined by the worst-case values of objectives in the initial population of solutions. The ideal hypervolume $HV_{\text{ideal}}$ is computed using this nadir point, along with an ideal point, which is approximated by running the underlying evolutionary algorithm one-time with a higher budget (e.g., doubled). It is worth noting that our reward function is instance-agnostic and thus applicable to different instances of varying sizes and complexities. We empirically observe that the reward function performs consistently well on different problems and delivers outstanding generalizability of trained models.

### 3.2 Graph-based policy learning and Training Algorithm

The agent, parameterized as a policy network, interacts with the environment by taking the current state as input, inferring an action, and collecting rewards based on the chosen action. The policy is then updated using the Proximal Policy Optimization (PPO) algorithm (Schulman et al., 2017) to train the parameterized policy. PPO is a widely used and highly effective policy gradient algorithm that utilizes a probability ratio between policies to maximize the improvement of the current policy without the risk of performance collapse. In our case, the agent utilizes a neural network that first processes the graph-based state representation through two Graph Convolutional Network (GCN) layers (Kipf & Welling, 2016). These layers are designed to extract and aggregate node embeddings (i.e., representations) effectively, capturing the essential structural information within the graph. Then, a global mean pooling operation is applied to average the node embeddings, producing a single embedding across the entire graph. The embedding is concatenated with an additional feature vector containing specific search budget information. The enriched embedding is finally fed into a linear layer to predict the mean values of action distributions. We have performed an ablation study in Appendix E, where we evaluate and test the setup and verify the effectiveness of our approach.

### 3.3 Multi-Objective Evolutionary Algorithm Deployment

Exact methods can achieve the accurate Pareto set in Multi-Objective Combinatorial Optimization (MOCO). However, the computational demands of these methods tend to increase exponentially with problem complexity, which often makes them impractical for large-scale applications. As a more feasible alternative, heuristic methods, particularly multi-objective evolutionary algorithms (MOEAs), are popular in practice due to their ability to effectively approximate Pareto fronts in a computationally efficient manner. In this work, we demonstrate GS-MODAC by applying it to two widely used algorithms: 1) NSGA-ii (Deb et al., 2002), which implements a non-dominated sorting mechanism with a crowding distance metric to preserve solution diversity throughout search, ensuring comprehensive exploration of the Pareto front; and 2) Multi-Objective Particle Swarm Optimization (MOPSO) (Coello & Lechuga, 2002), a swarm intelligence algorithm, which adjusts positions of particles by tracking both individual best locations and the best discoveries in the swarm. It integrates an archive to store non-dominated solutions to effectively cover the Pareto front.

## 4 Experiments

**Problems.** We apply our proposed method to two multi-objective combinatorial optimization problems: Flexible Job Shop Scheduling Problem (FJSP) and Capacitated Vehicle Routing Problem (CVRP). The FJSP involves scheduling multiple jobs, each composed of various operations, onto a set of machines. The operations of each job must be completed in a specific sequence, with each operation featuring a predefined processing time on specific machines. Based on the literature (Tamssaouet et al., 2022), we focus on minimizing Makespan, Balanced Workload, Average Flowtime, Total Workload, and Maximum Flowtime. We refer to the variants of FJSP as the Bi-, Tri- and Penta-FJPS, solving the first 2, first 3, and all 5 objectives, respectively. CVRP involves determining optimal routes for a fleet of vehicles to serve a set of customers. Each customer has a specific demand, and each vehicle has a capacity limit that must not be exceeded. The objectives are to minimize the total travel distances and the longest route. We refer to the CVRP problem composed

of these two objectives as the Bi-CVRP problem. Please refer to Appendix A for a comprehensive discussion of FJSP and CVRP, including the constraints and objectives addressed in this work.

**Instance generation.** For FJSP, we generate train and test instances for three distinct problem sizes: 1) 5 jobs and 5 machines (5j5m), 2) 10 jobs and 5 machines (10j5m), and 3) 25 jobs and 5 machines (25j5m), following the instance generation configuration of Song et al. (2022). We generate 200 instances for each problem configuration, consisting of 100 instances for training and 100 for testing. Each instance contains a varying number of operations per job, ranging from 4 to 8, and the processing time for each operation falls between 2 and 20 time units. The same instance sets are used for the experiments with 2, 3, or 5 objectives. For CVRP, we generate 3 distinct sizes of 100, 200, and 500 customers, according to the instance generation method in da Costa et al. (2021). We create 200 instances per problem size using random 2-dimensional coordinates for each customer and the depot in the 0 to 1 range. Each customer has a random demand between 1 and 9, and the vehicles have a capacity of 40 units.

**Baselines.** To show the performance of our proposed dynamic algorithm configuration method on solving multi-objective FJSP and CVPR, we use NSGAii as a base algorithm, whose values have been configured with rules of thumb, configuring the crossover parameter as 0.7 and the mutation parameter as 0.02 (Coello et al., 2007). Additionally, as shown in Appendix C, we empirically validate that our method effectively configures MOPSO, a swarm intelligence-based approach. We compare the proposed GS-MODAC against three algorithm configuration methods for tuning NS-GAii parameters: two widely used static AC methods, SMAC3 (Lindauer et al., 2022) and irace (López-Ibáñez et al., 2016), and a recent RL-based DAC approach, MADAC (Xue et al., 2022).

SMAC3 is a hyperparameter tuning method that combines Bayesian optimization and random forest regression. For the tuning, we use the generated test instances for each given instance size. Bayesian optimization is used to draw parameter configurations from the defined parameter configuration ranges and evaluate them on the provided tuning instances over 10.000 runs of the NSGAii configured algorithm, lasting between 5 to 14 hours for the Bi-CVRP instances and between 8 to 40 hours for the FJSP-variants. We also use the Iterated Race (irace) tuning method, which employs an iterative racing procedure. In each iteration (or 'race'), the worst-performing configurations are replaced with new ones, optimizing settings based on a set of given instances. irace was tuned with the same budget as the BO tuning method, taking between 3 and 12 hours for Bi-CVRP problem configurations and 5 to 20 hours for the FJSP-based variants, respectively. Since MADAC is designed to select discrete actions, we discretize the parameter space of NSGAii with 10 actions between 0.6 and 1.0 as crossover rate and between 0 and 0.1 for the mutation rate (in line with rules-of-thumb for EA parameter configurations (Coello et al., 2007)).

**Training.** We trained GS-MODAC for each problem configuration with randomly generated problem-instance sizes. The actions space for NSGAii is defined as two continuous actions with ranges $\langle 0.6, 1.0 \rangle$ and $\langle 0.0, 0.1 \rangle$ for the NSGAii crossover and mutation rates. The training process involved 1.000,000 steps for the scheduling problems and 2.500.000 steps for the routing, configured with 50 generations of search and a population size of 50. It was conducted on a Processor Intel(R) Core(TM) i7-6920HQ CPU @ 2.90GHz with 8.0GB of RAM and five parallel environments. The training duration varied for different-sized instance sets, taking around 11, 15, and 26 hours for the Bi-CVRP problem configurations and between 5 hours and 3 days for the different configured FJSP-based problems, where training on large instances with more objectives is more expensive. The training process spans 2000 epochs with 500 steps per epoch. The model parameters are set following Schulman et al. (2017), and network layers are configured with 64 nodes. The MADAC baseline model is trained according to Xue et al. (2022), taking between 2 and 8 hours for Bi-CVRP and 12 and 60 hours for the FJSP-based variants.

**Testing.** After training, the GS-MODAC agent is ready to be applied to tune the parameters of NS-GAii to solve unseen problem instances. Each experiment is performed by running each algorithm 10 times on 100 test instances for comparison. The evaluation is based on three metrics: average hypervolume (mean), best hypervolume (max), and standard deviation (std), which are computed by averaging all test instances for each problem. Hypervolumes are calculated using predefined reference points for each instance to ensure a fair comparison. The paper highlights the highest mean and max hypervolumes in bold and underlined values that significantly outperform all other methods using the Wilcoxon rank-sum test ($p < 0.05$).

Table 1: Performance comparison of different methods in solving 100 instances of various problems of varying sizes 10 times, based on the mean found hypervolume (mean), the best-found hypervolume (max), and the standard deviation (std).

| | Bi-FJSP - 5j5m | | | Bi-FJSP - 10j5m | | | Bi-FJSP - 25j5m | | |
|---|---|---|---|---|---|---|---|---|---|
| Method | mean | max | std | mean | max | std | mean | max | std |
| NSGAii | $1.87 \times 10^4$ | $2.02 \times 10^4$ | $1.21 \times 10^3$ | $3.82 \times 10^4$ | $4.11 \times 10^4$ | $2.29 \times 10^3$ | $9.41 \times 10^4$ | $9.93 \times 10^4$ | $4.84 \times 10^3$ |
| irace | $\mathbf{1.92 \times 10^4}$ | $\mathbf{2.04 \times 10^4}$ | $1.06 \times 10^3$ | $3.90 \times 10^4$ | $4.11 \times 10^4$ | $1.95 \times 10^3$ | $9.52 \times 10^4$ | $9.97 \times 10^4$ | $4.03 \times 10^3$ |
| SMAC3 | $1.91 \times 10^4$ | $2.04 \times 10^4$ | $1.09 \times 10^3$ | $3.89 \times 10^4$ | $4.13 \times 10^4$ | $2.19 \times 10^3$ | $9.51 \times 10^4$ | $9.97 \times 10^4$ | $4.46 \times 10^3$ |
| MADAC | $1.82 \times 10^4$ | $1.95 \times 10^4$ | $7.53 \times 10^2$ | $3.69 \times 10^4$ | $3.98 \times 10^4$ | $4.47 \times 10^3$ | $9.24 \times 10^4$ | $9.72 \times 10^4$ | $3.09 \times 10^3$ |
| GS-MODAC | $\mathbf{1.92 \times 10^4}$ | $\mathbf{2.04 \times 10^4}$ | $1.07 \times 10^3$ | $\mathbf{3.92 \times 10^4}$ | $\mathbf{4.15 \times 10^4}$ | $1.97 \times 10^3$ | $\mathbf{9.54 \times 10^4}$ | $\mathbf{10.0 \times 10^4}$ | $4.40 \times 10^3$ |

| | Tri-FJSP - 5j5m | | | Tri-FJSP - 10j5m | | | Tri-FJSP - 25j5m | | |
|---|---|---|---|---|---|---|---|---|---|
| Method | mean | max | std | mean | max | std | mean | max | std |
| NSGAii | $2.06 \times 10^6$ | $2.22 \times 10^6$ | $1.32 \times 10^5$ | $5.53 \times 10^6$ | $5.95 \times 10^6$ | $3.09 \times 10^5$ | $2.05 \times 10^7$ | $2.18 \times 10^7$ | $1.13 \times 10^6$ |
| irace | $\mathbf{2.11 \times 10^6}$ | $\mathbf{2.26 \times 10^6}$ | $1.16 \times 10^5$ | $5.47 \times 10^6$ | $5.82 \times 10^6$ | $2.65 \times 10^5$ | $2.07 \times 10^7$ | $2.20 \times 10^7$ | $1.07 \times 10^6$ |
| SMAC3 | $2.09 \times 10^6$ | $2.25 \times 10^6$ | $1.23 \times 10^5$ | $5.65 \times 10^6$ | $6.05 \times 10^6$ | $2.91 \times 10^5$ | $2.07 \times 10^7$ | $2.20 \times 10^7$ | $1.01 \times 10^6$ |
| MADAC | $1.99 \times 10^6$ | $2.14 \times 10^6$ | $8.86 \times 10^4$ | $5.39 \times 10^6$ | $5.87 \times 10^6$ | $5.97 \times 10^5$ | $2.09 \times 10^7$ | $2.20 \times 10^7$ | $1.97 \times 10^6$ |
| GS-MODAC | $2.10 \times 10^6$ | $2.25 \times 10^6$ | $1.16 \times 10^5$ | $\mathbf{5.70 \times 10^6}$ | $\mathbf{6.09 \times 10^6}$ | $2.99 \times 10^5$ | $\mathbf{2.14 \times 10^7}$ | $\mathbf{2.27 \times 10^7}$ | $1.09 \times 10^6$ |

| | Penta-FJSP - 5j5m | | | Penta-FJSP - 10j5m | | | Penta-FJSP - 25j5m | | |
|---|---|---|---|---|---|---|---|---|---|
| Method | mean | max | std | mean | max | std | mean | max | std |
| NSGAii | $6.01 \times 10^{10}$ | $6.48 \times 10^{10}$ | $3.70 \times 10^9$ | $3.96 \times 10^{11}$ | $4.31 \times 10^{11}$ | $2.42 \times 10^{10}$ | $5.08 \times 10^{12}$ | $5.48 \times 10^{12}$ | $2.75 \times 10^{11}$ |
| irace | $6.08 \times 10^{10}$ | $6.49 \times 10^{10}$ | $2.98 \times 10^9$ | $4.03 \times 10^{11}$ | $4.38 \times 10^{11}$ | $2.35 \times 10^{10}$ | $5.18 \times 10^{12}$ | $5.63 \times 10^{12}$ | $2.92 \times 10^{11}$ |
| SMAC3 | $6.08 \times 10^{10}$ | $6.50 \times 10^{10}$ | $3.29 \times 10^9$ | $3.97 \times 10^{11}$ | $4.29 \times 10^{11}$ | $2.24 \times 10^{10}$ | $4.95 \times 10^{12}$ | $5.33 \times 10^{12}$ | $2.71 \times 10^{11}$ |
| MADAC | $5.82 \times 10^{10}$ | $6.28 \times 10^{10}$ | $2.72 \times 10^9$ | $3.91 \times 10^{11}$ | $4.29 \times 10^{11}$ | $4.36 \times 10^{10}$ | $5.1 \times 10^{12}$ | $5.74 \times 10^{12}$ | $5.01 \times 10^{11}$ |
| GS-MODAC | $\mathbf{6.15 \times 10^{10}}$ | $\mathbf{6.58 \times 10^{10}}$ | $3.40 \times 10^9$ | $\mathbf{4.16 \times 10^{11}}$ | $\mathbf{4.52 \times 10^{11}}$ | $2.40 \times 10^{10}$ | $\mathbf{5.62 \times 10^{12}}$ | $\mathbf{6.07 \times 10^{12}}$ | $3.20 \times 10^{11}$ |

| | Bi-CVRP - 100 | | | Bi-CVRP - 200 | | | Bi-CVRP - 500 | | |
|---|---|---|---|---|---|---|---|---|---|
| Method | mean | max | std | mean | max | std | mean | max | std |
| NSGAii | $1.34 \times 10^2$ | $1.47 \times 10^2$ | 7.84 | $1.56 \times 10^2$ | $1.72 \times 10^2$ | 9.05 | $2.27 \times 10^2$ | $2.48 \times 10^2$ | $1.27 \times 10^1$ |
| irace | $1.34 \times 10^2$ | $1.48 \times 10^2$ | 8.02 | $1.57 \times 10^2$ | $1.72 \times 10^2$ | 9.53 | $2.27 \times 10^2$ | $2.48 \times 10^2$ | $1.26 \times 10^1$ |
| SMAC3 | $1.34 \times 10^2$ | $1.46 \times 10^2$ | 7.89 | $1.57 \times 10^2$ | $1.73 \times 10^2$ | 9.59 | $2.27 \times 10^2$ | $2.51 \times 10^2$ | $1.42 \times 10^1$ |
| MADAC | $\mathbf{1.35 \times 10^2}$ | $\mathbf{1.49 \times 10^2}$ | 8.01 | $\mathbf{1.61 \times 10^2}$ | $\mathbf{1.76 \times 10^2}$ | 9.22 | $2.33 \times 10^2$ | $2.54 \times 10^2$ | $1.29 \times 10^1$ |
| GS-MODAC | $\mathbf{1.35 \times 10^2}$ | $1.48 \times 10^2$ | 7.95 | $1.60 \times 10^2$ | $\mathbf{1.76 \times 10^2}$ | 9.50 | $\mathbf{2.35 \times 10^2}$ | $\mathbf{2.59 \times 10^2}$ | $1.41 \times 10^1$ |

## 4.1 EXPERIMENTAL RESULTS

We have formulated research questions to evaluate the performance of GS-MODAC. Specifically, these questions assess GS-MODAC's effectiveness compared to existing methods, its ability to generalize to previously unseen instances of varying sizes, its adaptability to more complex problem variants, and its scalability across different objectives.

**RQ1: How does GS-MODAC perform compared to the base algorithm NSGAii and three AC baseline methods for various problem types and sizes of objectives?**

Table 1 presents the performances of various methods, including the mean average performance, mean best-found solution, and the standard deviations for each method on two different problem types. The results highlight the effectiveness of GS-MODAC in controlling evolutionary parameters, achieving the best average and best-found solutions. For the smallest instance size in two-objective problems (Bi-), the baseline methods perform competitively, with the MADAC and irace configured baselines finding comparable mean and max solutions. However, GS-MODAC consistently excels in problem configurations with larger objective spaces, such as problems with more objectives and larger combinatorial search spaces (large instances configurations). This is particularly evident in the FJSP problem configurations with five objectives (Penta-), where GS-MODAC finds significantly better solutions than all baselines regarding mean and max found solutions. Specifically, for the Penta-FJSP problem configurations with 25 jobs and 5 machines, GS-MODAC's mean and maximum solutions are 8.2% and 5.7% better, respectively, than the best-performing baselines (irace and MADAC) and 10.6% and 10.8% better than the vanilla configured NSGAii method.

Additionally, Figures 2a and 2b illustrate that the GS-MODAC method converges significantly faster to find the optimal hypervolume for a Tri-FJSP with 10 jobs and 5 machines. It achieves a better-converged hypervolume, reaching superior minimum values for each objective, and is more widely spread across the different objective axes. Similar convergence patterns were observed for other instances, demonstrating the robustness of the GS-MODAC method. In Appendix D, we provide further analysis using alternative metrics to demonstrate GS-MODAC's ability to converge to the true Pareto front while maintaining a diverse set of high-quality solutions.

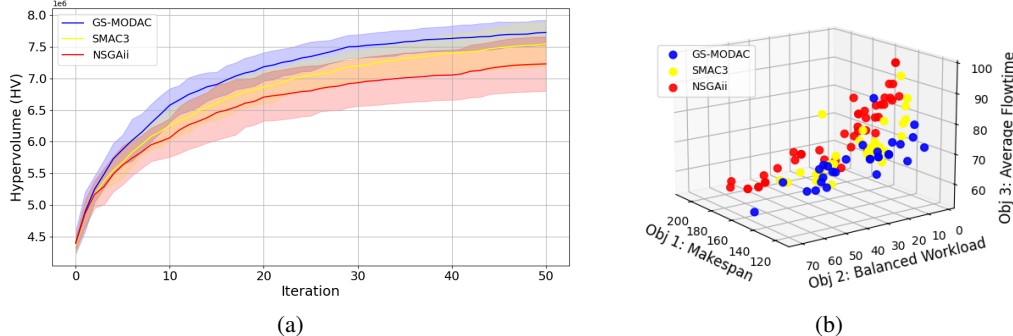

(a)  (b)

Figure 2: Comparison of GS-MODAC, SMAC3, and NSGAii solution methods: (a) Average convergence rates and (b) Pareto front distributions.

**RQ2: How well do the trained GS-MODAC models generalize to previously unseen instances of varying sizes?**

We assess the ability of the trained GS-MODAC models to solve previously unseen instances of different sizes. The results of this evaluation are presented in Table 2. The rows present the instance sizes on which the model is trained, and the columns show the instance sizes on which trained models are evaluated. We found that the models trained on smaller instances and deployed on larger instances experienced a slight decline in performance but still managed to achieve performance comparable to the best-performing baseline (MADAC) while outperforming the other baselines. Moreover, models trained on diverse instance sizes can effectively learn a robust, well-performing, all-around policy. The results suggest that our models could generalize and solve problem instances beyond the size on which they were trained.

Table 2: Generalizability of the trained models to solve unseen instances of different sizes.

| Method | Bi-CVRP - 100 | | | Bi-CVRP - 200 | | | Bi-CVRP - 500 | | |
|---|---|---|---|---|---|---|---|---|---|
| | mean | max | std | mean | max | std | mean | max | std |
| NSGAii | $1.34\times10^2$ | $1.47\times10^2$ | 7.84 | $1.56\times10^2$ | $1.72\times10^2$ | 9.05 | $2.27\times10^2$ | $2.48\times10^2$ | $1.27\times10^1$ |
| GS-MODAC - 100 | $\mathbf{1.35\times10^2}$ | $\mathbf{1.48\times10^2}$ | 7.95 | $1.59\times10^2$ | $1.75\times10^2$ | 9.25 | $2.32\times10^2$ | $2.55\times10^2$ | $1.31\times10^1$ |
| GS-MODAC - 200 | $\mathbf{1.35\times10^2}$ | $\mathbf{1.48\times10^2}$ | 8.22 | $\mathbf{1.60\times10^2}$ | $\mathbf{1.76\times10^2}$ | 9.50 | $2.33\times10^2$ | $2.56\times10^2$ | $1.36\times10^1$ |
| GS-MODAC - 500 | $1.33\times10^2$ | $1.47\times10^2$ | 8.52 | $\mathbf{1.60\times10^2}$ | $1.75\times10^2$ | 9.33 | $\mathbf{2.35\times10^2}$ | $\mathbf{2.59\times10^2}$ | $1.41\times10^1$ |
| GS-MODAC - all sizes | $1.34\times10^2$ | $\mathbf{1.48\times10^2}$ | 7.97 | $1.59\times10^2$ | $1.74\times10^2$ | 9.03 | $2.33\times10^2$ | $\mathbf{2.59\times10^2}$ | $1.41\times10^1$ |

**RQ3: How effectively can the trained GS-MODAC models adapt to solve previously unseen, more complex variants of problems?**  We assess the ability of the trained GS-MODAC models to solve previously unseen instances of two different, more complicated problem variants. These problems extend the Bi-, Tri- and Penta- objective FJSP with assembly constraints and sequence-dependent setup times, including additional precedence constraints between jobs and setup times operations on machines subject to the scheduling sequence. We test the proposed method on two variants of assembly scheduling, so-called 'DAFJS' and 'YFJS' scheduling problems as provided in Birgin et al. (2014), which have been extended with sequence-dependent setup times. The results, shown in Table 3, indicate that GS-MODAC trained on DAFJS-SDST demonstrates superior performance in most cases, except for the mean HV in the Penta-objective variant. Furthermore, the model configuration trained on the 10j5m problem variants effectively transfers to more complex problem scenarios. Notably, GS-MODAC trained on the 10j5m configurations outperforms all other baselines specifically tailored to the DAFJS and YFJS problem variants.

**RQ4: How effectively does the GS-MODAC model trained on a specific set of objectives adapt to different objectives than those encountered during training?**  We test the ability of the trained models to solve different variants of FJSP problems configured to optimize for objectives different from those explored in training. The problem we tested on (Bi-FJSP*) was configured to optimize for A and B, while the models were trained to optimize the C and D objectives, respectively. From table 4, it is clear that the trained models can be transferred to other configurations of the problem, finding solutions of similar or better quality than the configured baselines, with a similar performance gap as observed for two objective problem variants displayed in Table 1.

Table 3: Generalizability of trained models to solve instances of more complex problem variants.

| Method | Bi-DAFJS-SDST | | | Tri-DAFJS-SDST | | | Penta-DAFJS-SDST | | |
|---|---|---|---|---|---|---|---|---|---|
| | mean | max | std | mean | max | std | mean | max | std |
| NSGAii | $1.32\times10^6$ | $1.41\times10^6$ | $7.93\times10^4$ | $3.46\times10^8$ | $3.75\times10^8$ | $2.01\times10^7$ | $8.19\times10^{14}$ | $8.92\times10^{14}$ | $4.89\times10^{13}$ |
| irace | $1.41\times10^6$ | $1.47\times10^6$ | $5.60\times10^4$ | $3.49\times10^8$ | $3.75\times10^8$ | $1.86\times10^7$ | $8.10\times10^{14}$ | $8.76\times10^{14}$ | $3.97\times10^{13}$ |
| SMAC3 | $1.40\times10^6$ | $1.48\times10^6$ | $7.08\times10^4$ | $3.37\times10^8$ | $3.64\times10^8$ | $1.94\times10^7$ | $8.26\times10^{14}$ | $9.01\times10^{14}$ | $4.76\times10^{13}$ |
| GS-MODAC - FJSP-10j5m | $1.42\times10^6$ | $1.50\times10^6$ | $6.30\times10^4$ | $3.68\times10^8$ | $3.94\times10^8$ | $2.02\times10^7$ | $\mathbf{9.11\times10^{14}}$ | $9.93\times10^{14}$ | $5.48\times10^{13}$ |
| GS-MODAC - DAFJS-SDST | $\mathbf{1.43\times10^6}$ | $\mathbf{1.51\times10^6}$ | $6.48\times10^4$ | $\mathbf{3.73\times10^8}$ | $\mathbf{4.00\times10^8}$ | $2.25\times10^7$ | $9.05\times10^{14}$ | $\mathbf{1.00\times10^{15}}$ | $6.45\times10^{13}$ |
| Method | Bi-YFJS-SDST | | | Tri-YFJS-SDST | | | Penta-YFJS-SDST | | |
| | mean | max | std | mean | max | std | mean | max | std |
| NSGAii | $3.75\times10^6$ | $4.02\times10^6$ | $2.48\times10^5$ | $3.86\times10^9$ | $4.15\times10^9$ | $2.14\times10^8$ | $2.37\times10^{17}$ | $2.60\times10^{17}$ | $1.99\times10^{16}$ |
| irace | $4.02\times10^6$ | $4.24\times10^6$ | $1.78\times10^5$ | $3.87\times10^9$ | $4.13\times10^9$ | $1.98\times10^8$ | $2.38\times10^{17}$ | $2.60\times10^{17}$ | $2.98\times10^{16}$ |
| SMAC3 | $4.03\times10^6$ | $4.26\times10^6$ | $1.79\times10^5$ | $3.90\times10^9$ | $4.13\times10^9$ | $2.05\times10^8$ | $2.43\times10^{17}$ | $2.65\times10^{17}$ | $1.90\times10^{16}$ |
| GS-MODAC - FJSP-10j5m | $4.10\times10^6$ | $4.41\times10^6$ | $2.49\times10^5$ | $\mathbf{4.17\times10^9}$ | $4.53\times10^9$ | $2.64\times10^8$ | $\mathbf{2.69\times10^{17}}$ | $2.90\times10^{17}$ | $1.43\times10^{16}$ |
| GS-MODAC - YFJS-SDST | $\mathbf{4.20\times10^6}$ | $\mathbf{4.43\times10^6}$ | $2.00\times10^5$ | $\mathbf{4.17\times10^9}$ | $\mathbf{4.55\times10^9}$ | $3.48\times10^8$ | $2.65\times10^{17}$ | $\mathbf{2.95\times10^{17}}$ | $2.46\times10^{16}$ |

Table 4: Comparing the generalizability of the trained models to solve problem configuration to optimize different objectives that were not optimized in training.

| Method | Bi-FJSP* - 5j5m | | | Bi-FJSP - 10j5m | | | Bi-FJSP* - 25j5m | | |
|---|---|---|---|---|---|---|---|---|---|
| | mean | max | std | mean | max | std | mean | max | std |
| NSGAii | $3.49\times10^3$ | $3.57\times10^3$ | $4.05\times10^1$ | $9.64\times10^3$ | $9.88\times10^3$ | $1.31\times10^2$ | $\mathbf{3.93\times10^4}$ | $\mathbf{3.98\times10^4}$ | $2.57\times10^2$ |
| irace | $3.50\times10^3$ | $3.58\times10^3$ | $4.53\times10^1$ | $\mathbf{9.66\times10^3}$ | $9.91\times10^3$ | $1.48\times10^2$ | $3.92\times10^4$ | $3.97\times10^4$ | $2.62\times10^2$ |
| SMAC3 | $3.50\times10^3$ | $3.58\times10^3$ | $4.60\times10^1$ | $9.61\times10^3$ | $9.87\times10^3$ | $1.58\times10^2$ | $3.92\times10^4$ | $3.97\times10^4$ | $3.24\times10^2$ |
| GS-MODAC | $\mathbf{3.51\times10^3}$ | $\mathbf{3.58\times10^3}$ | $3.97\times10^1$ | $\mathbf{9.66\times10^3}$ | $\mathbf{9.93\times10^3}$ | $1.60\times10^2$ | $\mathbf{3.93\times10^4}$ | $\mathbf{3.98\times10^4}$ | $2.42\times10^2$ |

## 5 CONCLUSION

This paper presents a Graph-Supported Multi-Objective Dynamic Algorithm Configuration (GS-MODAC) method, leveraging a GNN and DRL to configure Evolutionary Algorithms for multi-objective combinatorial optimization problems dynamically. We model the state space by a graph to capture the convergence dynamics more effectively and propose an instance-agnostic reward function that is applicable to diverse problem types and sizes. Empirical results demonstrate that GS-MODAC outperforms traditional and DRL-based configuration methods, achieving better efficacy and adaptability. Additionally, it generalizes well to larger and more constrained problem instances not seen during training. In future work, we plan to explore advanced GNNs for Pareto front representations and apply specialized RL algorithms for contextual MDPs.

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

## A  TEST PROBLEM CONFIGURATIONS

The **Flexible Job Shop Scheduling Problem (FJSP)** is a popular scheduling problem where multiple jobs, each composed of several operations that must be completed in a specific order, must be scheduled to a set of machines. The problem contains a set of $n$ independent jobs $J = \{J_1, J_2, \ldots, J_n\}$ and $m$ independent machines $M = \{M_1, M_2, \ldots, M_m\}$, which together for an $n \times m$ Flexible Job Shop Scheduling Problem (FJSP). Each job $J_i$ consists operations $O_{i,j}$, where $O_{i,j}$ represents the $j$-th operation of the $i$-th job. These operations must be executed in sequence, meaning $O_{i,j+1}$ may only start after $O_{i,j}$ is completed. The processing time for operation $O_{i,j}$ on machine $M_k$ is denoted as $t_{i,j,k}$ and is known in advance. Each machine $M_k$ can process only one operation at a time, and operations cannot be interrupted (no preemption). The start and completion times for operation $O_{i,j}$ are denoted as $S_{i,j}$ and $C_{i,j}$ respectively, while $O_k$ is the set of operations assigned to on machine $M_k$.

This work focuses on five key minimization objectives commonly used in scheduling:

- Makespan: The total time required to complete all jobs, represented as $C_{\max} = \max_{i=1,\ldots,n} C_{i,j}$.
- Balance Workload: The disparity in workload distribution across machines, represented as $W_{\text{bal}} = W_{\max} - W_{\min}$, where $W_{\min} = \min_{k=1,\ldots,m} \sum_{(i,j) \in O_k} t_{i,j,k}$.
- Average flowtime: The average time duration jobs take from start to completion $F_{\text{avg}} = \frac{1}{n} \sum_{i=1}^{n} (C_{i,\text{last}} - S_{i,\text{first}})$.
- Total Workload: The cumulative sum of processing times for all jobs, defined as $W_{\text{total}} = \sum_{k=1}^{m} \sum_{(i,j) \in O_k} t_{i,j,k}$.
- Maximum flowtime: Denoting the longest time any job spends in the system from start to completion, defined as $F_{\max} = \max_{i=1,\ldots,n} (C_{i,\text{last}} - S_{i,\text{first}})$.

The **Capacitated Vehicle Routing Problem (CVRP)** is concerned with a fleet of vehicles that must deliver goods from a central depot to a set of customer locations while satisfying capacity constraints. The problem contains a set of $n$ customer locations $C = \{C_1, C_2, \ldots, C_n\}$, a depot location $C_0$, and $m$ identical vehicles. Each location $C_i$ has a demand $q_i$ representing the quantity of goods that need to be delivered to that particular customer. Each vehicle has a capacity of $Q$, representing the maximum total demand it can serve in a single route. Each vehicle $k$ can serve a demand of $\sum_{i=1}^{n} q_i \times y_{ik} \leq Q$, where $y_{ik}$ is a binary decision variable indicating whether vehicle $k$ serves customer $i$. The distance matrix $D$ is defined as $d_{ij}$, containing the distances between all pairs of locations, including customer locations and the depot, encapsulating the travel costs or distances associated with moving from one location to another.

The objectives considered in this work are to minimize the total distance traveled by all vehicles and the longest route:

- Total Travel Distance: $D_{\text{total}} = \sum_{k=1}^{m} \sum_{i=1}^{n} \sum_{j=1}^{n} d_{ij} \times x_{ijk}$
- Longest Route: $D_{\max} = \max_{k=1}^{m} \sum_{i=1}^{n} \sum_{j=1}^{n} d_{ij} \times x_{ijk}$

## B  MULTI-OBJECTIVE ALGORITHMS FOR MOCO

**NSGAii for FJSP.** To assess the efficacy of the proposed approach for FJSP, we devise a multi-objective Genetic Algorithm (GA) formulation inspired by Zhang et al. (2011). The solutions entail two integral components: Machine Selection and Operation Sequence. The first allocates operations to machines, while the second establishes the precedence of operations on the designated machines. Illustrated in Figure 3, a value of '4' in the initial position of Machine Selection indicates

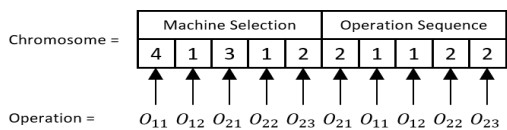

Figure 3: Chromosome Representation FJSP MOGA (Zhang et al., 2011)

the scheduling of operation $O_{1,1}$ on the fourth machine alternative. Subsequently, the Operation Sequence component arranges this operation as second, after $O_{2,1}$.

The population is initialized using Global, Local, and Random Methods. Global Method assigns operations to machines sequentially, minimizing the total processing times of individual machines. Local Method minimizes the max machine processing times for individual jobs. Random Method allocates operations to machines randomly. The Operation Sequence is initialized randomly for all methods. Crossover is applied to the Machine Selection component using two-point and uniform crossover while precedence-preserving order-based crossover (POX) is applied to the Operation Sequence component. POX preserves relative scheduling positions for a randomly selected set of jobs and reschedules the remaining operations according to the other crossovered individual solution. We generate 60% of the initial population using Global Method, 30% using Local Method, and 10% using Random Method. Machine Selection crossovers are in 50% two-point and 50% uniform crossover. To solve the multi-objective FJSP variant using the GA formulation from Zhang et al. (2011), we employ Non-dominated Sorting Genetic Algorithm-II (NSGA-II) for selection (Deb et al., 2002).

**NSGAii for CVRP.** Subsequently, we apply a multi-objective Genetic Algorithm (GA) formulation to assess the efficacy of the proposed approach for CVRP. The solutions are initialized with random routes, where each solution is represented as a list of values corresponding to the sequence in which customers are visited in the CVRP.

The selected parents undergo crossover and mutation to produce offspring, using ordered crossover and a shuffle mutation; crossoign over two segments from two selected parent solutions, and randomly swapping elements within solutions with a given probability. The next generation is formed by selecting individuals from the combined population based on their rank (front) and crowding distance. The algorithm prioritizes individuals from lower fronts and those with higher crowding distances to ensure a diverse and high-quality population. Non-dominated Sorting Genetic Algorithm-II (NSGA-II) is applied for the selection (Deb et al., 2002).

**MOPSO for CVRP.** We define a Multi-Objective Particle Swarm Optimization (MOPSO) algorithm for the Capacitated Vehicle Routing Problem (CVRP). In this algorithm, solutions (particles) are initialized with random routes, represented as a list of random values where each value corresponds to a customer in the CVRP. Each particle also has associated velocities that represent changes in these routes. The initial fitness values for each particle are calculated by sorting the customers based on the values in the particle's position to determine the routes. Each particle's personal best solution is recorded, and all the best-found solutions are stored in a separate list.

In each generation of the search, the positions and velocities of the particles are updated based on their personal best and a global best chosen from the Pareto front (randomly selected when multiple best solutions are available). The velocity update formula incorporates cognitive coefficients ($\phi 1$), social coefficients ($\phi 2$), and an inertia weight. Initially, random coefficients (u1 and u2) are generated for each particle dimension to balance exploration and exploitation. The velocity update consists of two components: one influenced by the particle's personal best and the other by the global best from the Pareto front. The velocity for each particle dimension is calculated using these components, scaled by the respective random coefficients and adjusted by the inertia weight. The updated velocity is clamped within predefined minimum (min) and maximum (max) bounds to remain within valid bounds. The particle's new position is determined by adding the updated velocity to the current position. Finally, each position is clamped to remain within valid bounds, typically between 0 and 1, ensuring the particle stays within the feasible solution space.

After the update, the fitness of the particles is evaluated. The particles' personal bests and the list of best solutions are updated using non-dominated sorting to retrieve Pareto-optimal solutions. A

selection mechanism based on Pareto dominance (using NSGA-II) is applied to maintain a diverse and optimal set of solutions in the population. For this work, we configure the vanilla MOPSO algorithm for CVRP with the following parameters: social and cognitive coefficients are configured as 2.0, an inertia weight of 0.9, and we use a population size of 50 particles for 50 generations. GS-MODAC is configured to tune the social and cognitive coefficients between 1 and 3 and the inertia weight factor between 0.6 and 0.9.

## C  ALTERNATIVE MOEA RESULTS

We show another instantiation of the proposed GS-MODAC method, where the DRL agent dynamically configures the parameters of a multi-objective PSO (MOPSO) algorithm. Table 5 shows GS-MODAC can effectively improve the performance of MOPSO, achieving better results on solving the two-objective CVRP problems with sizes 20, 50, and 100.

Table 5: Performance comparison of the proposed method for dynamic algorithm configuration of Multi-Objective Particle Swarm Optimization (MOPSO) Algorithm.

|  | Bi-CVRP - 20 | | |
|---|---|---|---|
| Method: | mean | max | std |
| MOPSO | $3.21 \times 10^1$ | $3.75 \times 10^1$ | 3.47 |
| GS-MODAC | $\mathbf{3.28 \times 10^1}$ | $\mathbf{3.77 \times 10^1}$ | **3.20** |
|  | Bi-CVRP - 50 | | |
| Method | mean | max | std |
| MOPSO | $5.82 \times 10^1$ | $6.90 \times 10^1$ | **5.65** |
| GS-MODAC | $\mathbf{6.27 \times 10^1}$ | $\mathbf{8.08 \times 10^1}$ | 9.73 |
|  | Bi-CVRP - 100 | | |
| Method | mean | max | std |
| MOPSO | $8.67 \times 10^1$ | $9.75 \times 10^1$ | **5.87** |
| GS-MODAC | $\mathbf{1.06 \times 10^2}$ | $\mathbf{1.46 \times 10^2}$ | $2.42 \times 10^1$ |

## D  ALTERNATIVE PERFORMANCE METRICS

We further evaluate performances using additional metrics commonly employed in multi-objective optimization research: Inverted Generational Distance (IGD), Inverted Generational Distance Plus (IGD+), and the number of non-dominated solutions. These results are gathered using the same setup as in the paper for the J25m5 scheduling problem with 2,3 and 5 objectives. The results, shown in Table 6, highlight the effectiveness of GS-MODAC, as it finds a significantly higher number of "best" solutions (max) and achieves lower IGD+ values.

In terms of IGD, GS-MODAC outperforms the baseline methods in the experiments with more objectives. It is important to note that while IGD provides valuable insights into the proximity of solutions to the Pareto front, it is sensitive to the distribution of solutions and more subject to outliers. In contrast, IGD+ is less sensitive to these factors, making it a more reliable measure for evaluating the overall quality and diversity of solutions. Therefore, the consistently lower IGD+ values across multiple objectives achieved by GS-MODAC highlight its ability to converge to the true Pareto front while maintaining a diverse set of high-quality solutions.

## E  ABLATION STUDY

An ablation study was conducted to account for the performance of the different components of the proposed method. As a first ablation, we trained GS-MODAC without the additional feature vector containing the normalized remaining search budget. Table 7 shows that, without this vector, the performance of the proposed method decreased on average with 0.8%, 3.2%, and 1.7%, respectively, for 2, 3, and 5 objectives for solving the scheduling problem with the 25j5m instances. Another ablation was conducted with only one GCN layer. This resulted in an average performance decrease of 1.7%, 3.2%, and 3.4% for 2, 3 and 5 objectives.

Table 6: Additional Performance Metrics: Inverted Generational Distance (IGD), Inverted Generational Distance Plus (IGD+), and nr. of non-dominated solutions.

| Bi-FJSP - 25j5m | | | | | | | | |
|---|---|---|---|---|---|---|---|---|
| | IGD | | | IGD+ | | | non-dominated solutions | | |
| | mean | min | std | mean | min | std | mean | max | std |
| NSGAii | 19.07 | 8.47 | 11.49 | 15.79 | 4.05 | 12.25 | 5.06 | 8.78 | 2.11 |
| irace | 15.66 | 7.23 | 8.41 | 11.23 | 2.52 | 9.10 | 4.92 | 8.07 | **1.94** |
| SMAC3 | **15.13** | **6.99** | **7.75** | 15.13 | 6.99 | **7.75** | 4.99 | 8.26 | 1.98 |
| GS-MODAC | 15.77 | 7.41 | 9.10 | **9.82** | **1.25** | 10.10 | **6.81** | **20.69** | 5.80 |

| Tri-FJSP - 25j5m | | | | | | | | |
|---|---|---|---|---|---|---|---|---|
| | IGD | | | IGD+ | | | non-dominated solutions | | |
| | mean | min | std | mean | min | std | mean | max | std |
| NSGAii | 22.20 | 15.77 | **6.00** | 18.09 | 10.11 | **6.78** | 36.29 | 54.10 | 10.18 |
| irace | 19.23 | 13.44 | 6.09 | 12.29 | 5.36 | 6.83 | 35.30 | 51.29 | 10.06 |
| SMAC3 | **19.19** | **13.07** | 6.13 | 11.24 | 3.83 | 7.02 | 35.25 | 52.95 | 9.81 |
| GS-MODAC | 20.63 | 13.86 | 6.77 | **8.40** | **2.46** | 6.79 | **36.64** | **54.83** | **11.07** |

| Penta-FJSP - 25j5m | | | | | | | | |
|---|---|---|---|---|---|---|---|---|
| | IGD | | | IGD+ | | | non-dominated solutions | | |
| | mean | min | std | mean | min | std | mean | max | std |
| NSGAii | 28.58 | 23.42 | 4.19 | 21.18 | 14.18 | **4.70** | 172.20 | 223.14 | 32.46 |
| irace | 23.97 | 19.88 | **4.09** | 13.67 | 7.38 | 4.78 | 203.94 | 263.34 | 40.22 |
| SMAC3 | 26.10 | 21.16 | 4.46 | 17.17 | 9.91 | 5.22 | 185.93 | 243.54 | **35.94** |
| GS-MODAC | **23.82** | **19.08** | 5.06 | **8.71** | **3.25** | 5.03 | **231.13** | **311.21** | 52.14 |

Table 7: Ablation study, comparing GS-MODAC configured without additional feature vector and with one configured GCN layer.

| | Bi-FJSP - 25j5m | | | Tri-FJSP - 25j5m | | | Penta-FJSP - 25j5m | | |
|---|---|---|---|---|---|---|---|---|---|
| | mean | max | std | mean | max | std | mean | max | std |
| MADAC | $9.24 \times 10^4$ | $9.72 \times 10^4$ | **$3.09 \times 10^3$** | $2.09 \times 10^7$ | $2.20 \times 10^7$ | $1.97 \times 10^6$ | $5.1 \times 10^{12}$ | $5.74 \times 10^{12}$ | $5.01 \times 10^{11}$ |
| GS-MODAC (No feature) | $9.47 \times 10^4$ | $9.92 \times 10^4$ | $4.21 \times 10^3$ | $2.07 \times 10^7$ | $2.21 \times 10^7$ | $1.10 \times 10^6$ | $5.53 \times 10^{12}$ | $6.02 \times 10^{12}$ | $3.42 \times 10^{11}$ |
| GS-MODAC (One GCN) | $9.38 \times 10^4$ | $9.88 \times 10^4$ | $4.87 \times 10^3$ | $2.07 \times 10^7$ | $2.19 \times 10^7$ | **$1.02 \times 10^6$** | $5.49 \times 10^{12}$ | $5.98 \times 10^{12}$ | $3.37 \times 10^{11}$ |
| GS-MODAC | **$9.54 \times 10^4$** | **$10.0 \times 10^4$** | $4.40 \times 10^3$ | **$2.14 \times 10^7$** | **$2.27 \times 10^7$** | $1.09 \times 10^6$ | **$5.62 \times 10^{12}$** | **$6.07 \times 10^{12}$** | **$3.20 \times 10^{11}$** |

In addition, we adapted GS-MODAC for the Penta-FJSP - 25j5m problem by replacing the GCN layers with Transformers and Graph Attention Networks (GAT). The results presented in Table 8 indicate that Transformers are a viable alternative, with average performance being only 0.3% lower than GCN and its best-found solutions only 0.5% worse. The performance difference of GAT layers is more substantial, with an average degradation of 1.4%.

Table 8: Comparison of different network architectures (GCN, Transformer, GAT) for GS-MODAC.

| | Penta-FJSP - 25j5m | | |
|---|---|---|---|
| | mean | max | std |
| GCN | **$5.62 \times 10^{12}$** | **$6.07 \times 10^{12}$** | **$3.20 \times 10^{11}$** |
| transformer | $5.61 \times 10^{12}$ | $6.04 \times 10^{12}$ | $3.60 \times 10^{11}$ |
| GAT | $5.54 \times 10^{12}$ | $6.04 \times 10^{12}$ | $3.35 \times 10^{11}$ |

# F COMPARISON TO END-TO-END METHOD P-MOCO

We compare GS-MODAC with P-MOCO (Lin et al., 2022), a commonly used learning-based approach for Pareto set learning, and NHDE-P (Chen et al., 2024), a recent enhancement to P-MOCO that incorporates neural heuristics with diversity enhancement (NHDE). NHDE-P leverages graph attention mechanisms to capture relationships between the instance graph and the Pareto front, providing improved guidance for methods like P-MOCO. It is important to note that P-MOCO and NHDE-P feature a specialized network structure tailored to simple TSP and CVRP. Therefore, they cannot address scheduling problems such as FJSP. Hence, we compare with these methods to solve CVRP with 2 objectives. We followed the training details provided in Lin et al. (2022) and Chen et al. (2024) for P-MOCO and NHDE-P and trained all methods on the same set of instances of size

100. Their performance was evaluated according to the setup outlined in Section 4, using the same instances and reference points. The results, shown in Table 9, compare the best HV values obtained, in alignment with the setup from Lin et al. (2022) and Chen et al. (2024).

Table 9: Comparison of Hypervolume (HV) values achieved by NSGAii, and by P-MOCO, NHDE-P, and GS-MODAC (all trained on size 100) for Bi-CVRP instances of varying sizes.

|  | Bi-CVRP - 20 | Bi-CVRP - 50 | Bi-CVRP - 100 |
|---|---|---|---|
| NSGAii | 42.86 | 151.24 | 363.87 |
| P-MOCO | 34.71 | 152.83 | 438.06 |
| NHDE-P | 41.31 | **152.97** | **446.13** |
| GS-MODAC | **45.18** | 152.87 | 366.63 |

The results indicate that both P-MOCO and NHDE-P perform better than GS-MODAC when trained and tested on instances of size 100, which is expected since both methods learn policies tailored to specific instances. However, in terms of generalizability, P-MOCO is inferior to GS-MODAC, as seen in the performances on Bi-CVRP-20 and Bi-CVRP-50. This indicates that GS-MODAC has significantly better generalization capability than P-MOCO, which is somewhat overfitted to a specific size used in training. NHDE-P shows stronger performance and generalizability capabilities than P-MOCO, yet falls short when tested on the smallest instance size. Additionally, we also observe that GS-MODAC outperforms NSGAii when generalizing to different sizes.

To further assess robustness, we tested models trained on size-100 instances against instances generated from a normal distribution (mean 0.3, standard deviation 0.1) with 5% outliers, differing from the uniform distribution used for training. The results demonstrate that GS-MODAC consistently outperforms P-MOCO across all sizes, indicating its superior generalization capability. Unlike P-MOCO, which tends to overfit not only to a specific problem size but also to the distribution of training instances, GS-MODAC shows robust performance across different instance distributions. Although NHDE-P achieves the best results for the largest instance configuration, its performance on smaller instances, while better than P-MOCO, falls short compared to GS-MODAC. Additionally, GS-MODAC keeps surpassing NSGAii in the generalization to various instance distributions and sizes.

Table 10: Performances in terms of Hypervolume (HV) on Bi-CVRP instances with different distributions and outliers, compared to the training instances.

|  | Bi-CVRP - 20 | Bi-CVRP - 50 | Bi-CVRP - 100 |
|---|---|---|---|
| NSGAii | 59.34 | 192.64 | 455.80 |
| P-MOCO | 51.05 | 186.35 | 454.76 |
| NHDE-P | 57.75 | 192.20 | **462.91** |
| GS-MODAC | **59.55** | **194.47** | 458.46 |

## G  ALTERNATIVE SOLUTION CRITERIA IN REWARD FUNCTION

We have updated our reward function to optimize for Inverted Generational Distance (IGD) instead of hypervolume. Apart from the fact that IGD is a minimization objective, the reward function's structure remains unchanged. We set $IGD_{ideal}$ as 0 and calculate IGD using an approximated Pareto front generated through a single GA search with double the usual search budget. The results, shown in Table 11, demonstrate the effectiveness of GS-MODAC with this IGD function for various performance metrics. Specifically, it consistently outperforms NSGAii. However, compared to the original HV-based reward function, we do not observe significant improvement in performance.

## H  OBJECTIVES GENERALIZABILITY OF GS-MODAC

We assessed GS-MODAC's ability to transfer knowledge from objectives A/B to C/D. We argue this capability stems from the algorithm's capacity to capture and generalize patterns in the graph state space, leveraging latent structural or topological similarities. To understand what GS-MODAC has learned, we compared the graph state spaces for objectives A/B and C/D in Figure 4. The figure

Table 11: Reward Function Comparison: GS-MODAC using hypervolume-based (GS-MODAC (HV)) and Inverted Generation Distance (GS-MODAC (IGD)) rewarding.

| | Bi-FJSP - 25j5m | | | | | |
| | HV mean | HV max | HV std | IGD mean | IGD min | IGD std |
|---|---|---|---|---|---|---|
| NSGAii | $9.41 \times 10^4$ | $9.93 \times 10^4$ | $4.84 \times 10^3$ | 19.07 | 8.47 | 11.49 |
| GS-MODAC (HV) | $\mathbf{9.54 \times 10^4}$ | $\mathbf{10.0 \times 10^4}$ | $\mathbf{4.40 \times 10^3}$ | **15.77** | **7.41** | 9.10 |
| GS-MODAC (IGD) | $9.48 \times 10^4$ | $9.94 \times 10^4$ | $4.26 \times 10^3$ | 16.07 | 7.57 | **8.90** |
| | Tri-FJSP - 25j5m | | | | | |
| | HV mean | HV max | HV std | IGD mean | IGD min | IGD std |
| NSGAii | $2.05 \times 10^7$ | $2.18 \times 10^7$ | $1.13 \times 10^6$ | 22.20 | 15.77 | 6.00 |
| GS-MODAC (HV) | $\mathbf{2.14 \times 10^7}$ | $\mathbf{2.27 \times 10^7}$ | $\mathbf{1.09 \times 10^6}$ | 20.63 | 13.86 | 6.77 |
| GS-MODAC (IGD) | $2.13 \times 10^7$ | $2.26 \times 10^7$ | $1.94 \times 10^6$ | **20.49** | **13.59** | **6.37** |
| | Penta-FJSP - 25j5m | | | | | |
| | HV mean | HV max | HV std | IGD mean | IGD min | IGD std |
| NSGAii | $5.08 \times 10^{12}$ | $5.48 \times 10^{12}$ | $2.75 \times 10^{11}$ | 28.58 | 23.42 | 4.19 |
| GS-MODAC (HV) | $5.62 \times 10^{12}$ | $6.07 \times 10^{12}$ | $\mathbf{3.20 \times 10^{11}}$ | **23.82** | **19.08** | 5.06 |
| GS-MODAC (IGD) | $\mathbf{5.65 \times 10^{12}}$ | $\mathbf{6.10 \times 10^{12}}$ | $3.37 \times 10^{11}$ | 24.10 | 19.32 | **5.04** |

illustrates the states and actions over time (every 10 iterations) of a model trained on A/B when applied to solve the same problem instance under both objective configurations.

From the figure, it can be observed that the graph representations for A/B and C/D share visual similarities but also reveal distinct differences. Both configurations exhibit similar patterns during the search: solutions are initially scattered across the objective space at iteration 0 and converge toward the bottom-left corner. Note that the normalization applied to generate these graphs depends solely on the min and max bounds obtained during the search at that point, with no future information or approximated objectives used. However, we observe that the convergence pattern differs. For A/B, the algorithm converges faster toward solutions that perform well for both objectives. For C/D, solutions are more dispersed across objective scales, reflecting the competing nature of the objectives. Action selection also varies between configurations. For A/B, the model favors lower crossover rates and relies more on mutation, while for C/D, it employs higher crossover rates to enable more extensive exploration. The graph states can explain this: A/Bs solutions that are more similar in objective values seem to benefit from more local exploration, while C/D's competing objectives demand broader exploration. This indicates GS-MODAC's ability to adapt its strategy based on the specific characteristics of the state of the search highlighted in the graph despite being trained on the same problem with different configured objectives.

# I    COMPLEXITY ANALYSIS

We profiled GS-MODAC to assess its computational complexity, focusing on graph state configuration and policy network inference. Results show the actor's inference time is 0.13 seconds, and state extraction takes 0.2 seconds, together accounting for 2.0% of the total time for the smallest scheduling problem instances. For larger problems, this proportion decreases significantly as solution evaluations dominate computation. Despite a slight overhead, its substantial performance gains justify GS-MODAC's minimal additional cost.

| | Bi-FJSP - 5j5m | Penta-FJSP - 5j5m | Bi-FJSP - 25j5m | Penta-FJSP - 25j5m |
|---|---|---|---|---|
| Total Inference Time | 15.09s | 15.46s | 305s | 302s |
| Total State Configuration Time | 0.18s | 0.21s | 0.23s | 0.22s |
| Total Policy Inference Time | 0.12s | 0.12s | 0.14s | 0.13s |

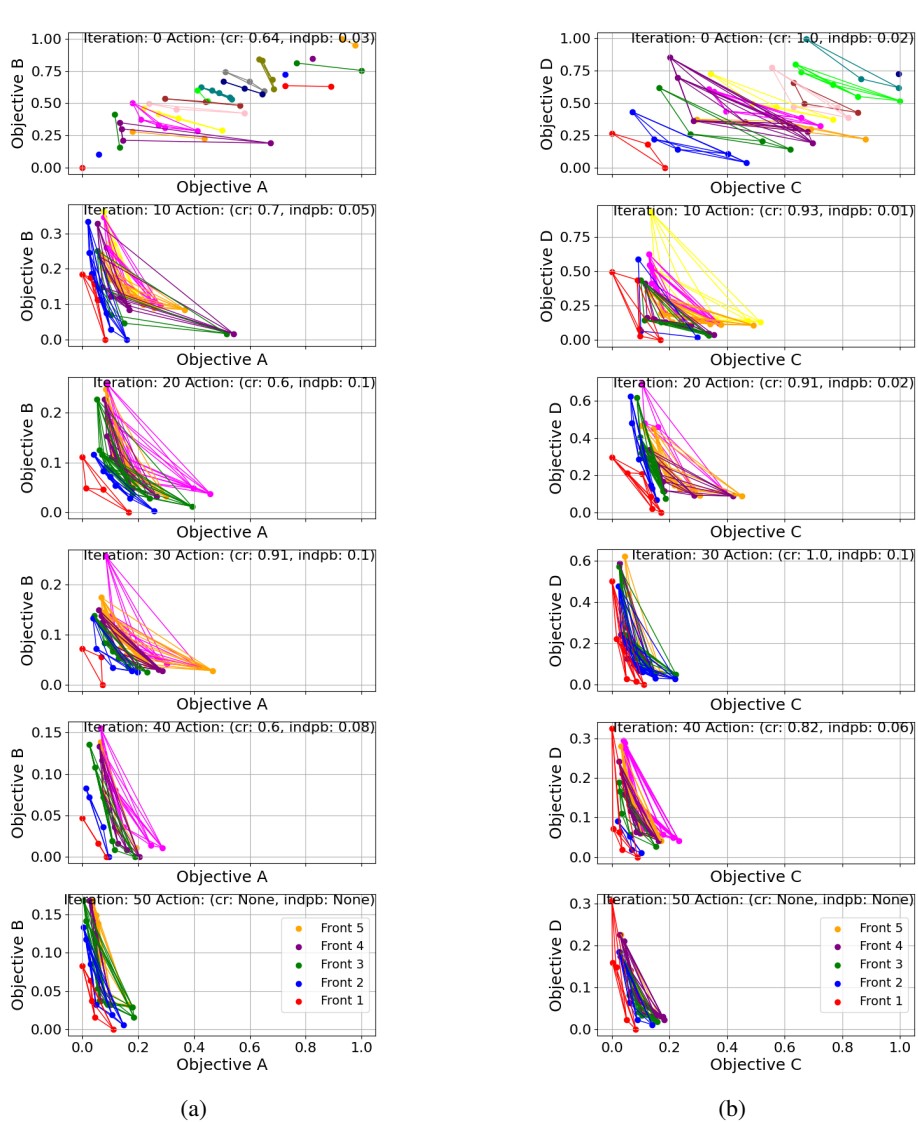

(a)                                     (b)

Figure 4: Comparison of different state patterns and the actions sampled by GS-MODAC, trained on objectives A/B, when deployed to: (a) objectives A/B and (b) objective C/D.

