# OpenReview forum: "Graph-Supported Dynamic Algorithm Configuration for Multi-Objective Combinatorial Optimization"
_ICLR.cc/2025/Conference — Submitted to ICLR 2025_

### Official Review · Reviewer_A1gs · 2024-10-17

**Soundness:** 2
**Presentation:** 3
**Contribution:** 2
**Rating:** 5
**Confidence:** 5

**Summary:**

This paper introduces a DRL-based algorithm configuration framework (GS-MODAC) for addressing multi-objective COPs. The authors propose using GNN as a feature extraction mechanism for capturing the optimisation dynamics. They then formulate the optimisation process of COPs as MDP, where the state is the GNN-based pareto feature, the action is cossing the hyper-parameter values for the multi-objective algorithm, and the reward is a tailor one with capability of assigning more credits for convergence situations to reinforce the learning. The experiments are facilitated to validate the proposed method on several representative COPs, including the in-distribution generalization and out-of-distribution generalization. Under the given experimental settings, GS-MODAC achieves competitive results compared to several baselines, including SMAC and MADAC,

**Strengths:**

Overall, the writing is reader-friendly.

**Weaknesses:**

Several major weaknesses hinder this paper to get the acceptance boardline:

a) Limited review on related works: In the last two years, many related works that leverage RL/DRL for algorithm configuration in EAs or SI algorithms are proposed. However, this paper only reviews the related works before 2022. Even for papers before 2022, there are already realted works investigating how to configure multi-obejctive optimizers through learning system, for instance:

[1] Ning W et al. (2018). https://link.springer.com/article/10.1007/s13748-018-0155-7.

[2] Huang et al. (2020).  https://www.sciencedirect.com/science/article/abs/pii/S1568494620306311.

[3] Tian et al.  (2023). https://ieeexplore.ieee.org/abstract/document/9712324.

there are more.

b)  Limited novelty: The paradigm in this paper show certain similarity with these related works despite the tartget optmisation problems and the GNN-based feature extraction. However this novelty is also not 'novel' enough considering a recent NuerIPS paper (Chen et al (2023). https://arxiv.org/abs/2310.15195) , where graph attention is used for autonomous pareto feature extraction. If the authors claim this novelty, an in-depth analysis should be carried out to compare at least the above usage of graph information.

c)  Unclear elaboration of methodology: what exactly is the node embedding and the final state representation?

d)  Overpromise in adaptability: the authors claim that: 'prospective applicability to different evolutionary algorithms'. However, the action spaces in different multi-objective optimizers can vary. This means the proposed network structure is not generic to handle this. More importantly, the authors claim than: 'We trained GS-MODAC for each problem configuration with randomly generated
problem-instance sizes'. I am confused here since an advantage obervaed in recent learning-assisted EC works is that the policy could be trained on all problem configurations to achieve maximum generalization. Can a GS-MODAC model trained on FJSP 5j5m generalize well on other configurations? Can GS-MODAC be trained on all isntances of all configurations and then generalize well?

e) Limited in-depth analysis: the experimental results revolve around the performance, this leads to the limited interpretability of this work. For instance, in RQ 4, can authors explain why GS-MODAC can transfer the knowledge learned on C/D objectives to A/B objectivess. Is this due to the similarity among the objectives? More importantly, what has GS-MODAC learned? I suggest two improvements on this point: i) visualization of the learned graph scores to analyse the learned pareto pattern. ii) analyse the relationship between different pareto patterns and the action distributions controlled by GS-MODAC.

f) Minors: when we talk about the algorithms in EC domain, evolutionary algorithms (EAs) include GA, DE, ES etc., swarm intelligence algorithms (SI) include PSO, ACO etc. Such definition should be made clear. (NSGAII->EAs, MOPSO->SI)

**Questions:**

See weaknesses.

---

> ### Author Response · Authors · 2024-11-23
>
> We thank the reviewer for their time and constructive feedback. Our responses are below, with the revised paper attached as a PDF.
>
> **W1: Related works**
>
> We included the mentioned papers and other related works in the related work section (in red). Most RL-based approaches for multi-objective optimization rely on carefully selected convergence and fitness landscape features. In contrast, our approach uses a graph representation to learn advanced features during training, eliminating the need for feature design. While many methods focus on operator selection, they overlook configuring operator parameters. Unlike these approaches, often aimed at continuous optimization, we focus on multi-objective combinatorial optimization, where the discrete, high-dimensional, non-smooth solution spaces make generalizing parameter configurations across instances challenging.
>
> **W2: Graph information comparison**
>
> Chen et al. introduce NHDE, using graph attention to model relationships between the instance graph and the first Pareto front to guide methods like PMOCO. However, similar to decomposition-based methods, its applicability is limited by the specialized network structure for specific problems like TSP and CVRP with distance-based objectives. In contrast, our method is more versatile, solving different problem categories (routing and scheduling) with multiple objectives, including problems that are difficult to represent in a graph for decomposition or end-to-end approaches (e.g., DAFJS-STST).
>
> Unlike NHDE, our graph representation captures the state of the evolutionary search. Rather than focusing on encoded solutions and solely the first Pareto front, we represent the entire objective space of all solutions in the population, mapping all fronts and normalizing them based on objective convergence. To demonstrate our method’s performance, we include NHDE-P (NHDE with PMOCO) in Appendix F. While inferior in size 100, GS-MODAC is superior for generalizing to different distributed instances in the other sizes.
>
> **W3: Unclear methodology**
>
> Nodes represent the objective values of solutions at a given iteration, mapped onto multiple objective planes and normalized based on objective convergence. The graph interconnects the points in the different Pareto fronts. As such, the final state represents a structured visualization of the final solution space and the convergence of the method. We have added this clarification to the caption of Figure 1 and extended our motivation in the introduction.
>
> **W4-1: Action space**
>
> To apply the method to different algorithms, only the output layer needs adjustment to match the number of parameters. Appendix B shows its application to MOPSO, which requires more parameters than NSGA-II. The results in Table 5 (Appendix C) demonstrate that GS-MODAC improves MOPSO's performance, highlighting its effectiveness across algorithms.
>
> **W4-2: Model generalization**
>
> The models can generalize to other configurations, and the method can train a well-performing policy when trained on all sizes. To show this, we used the 5j5m and 10j5m models for larger Tri-FJSP instances and trained GS-MODAC on all sizes. The results show that the 10j5m-trained model scales well to larger configurations, while the 5j5m model shows lower scalability. The model trained on all sizes performs well overall, achieving the best 5j5m results and second-best for 10j5m, performing comparably to the best baseline.
> ||**5j5m** ||**10j5m**||**25j5m**||
> |:-:|:-:|:-:|:-:|:-:|:-:|:-:|
> ||HV mean|HV max|HV mean|HV max|HV mean|HV max
> GS-MODAC - 5j5m|2.10×10^6|**2.25×10^6**|5.51×10^6|5.91×10^6|2.03×10^7|2.16×10^7
> GS-MODAC - 10j5m|2.09×10^6|**2.25×10^6**|**5.70×10^6**|**6.09×10^6**|2.99×10^5|2.13×10^7
> GS-MODAC (all)|**2.11×10^6**|2.23×10^6|5.63×10^6|6.05×10^6|2.09×10^7|2.23×10^7
> GS-MODAC|2.10×10^6|**2.25×10^6**|**5.70×10^6**|**6.09×10^6**|**2.14×10^7**|**2.27×10^7**
>
> **W5: In-depth analysis**
>
> The method is not provided with information about encoded solutions or the objective to be optimized, relying entirely on generalizing graph states to guide the underlying algorithm. To analyze this, we visualized the relationship between the states and actions in Appendix G using a model trained on objectives A/B to solve both objective configurations.
> While the graph patterns are visually similar, A/B converges faster toward balanced solutions, while C/D obtains solutions with more competing objectives, resulting in wider Pareto fronts. This leads to different actions: A/B is configured with high mutation and lower crossover rates, while C/D is configured with high crossover rates. The graph states can explain this: A/Bs solutions that are more similar in objective values seem to benefit from more local exploration, while C/D's competing objectives demand broader exploration. This indicates the method’s ability to adapt actions to the graph's features.
>
> **W6: Minors:**
>
> Thanks for these pointers, we have updated this accordingly.

---

> ### Comment · Reviewer_A1gs · 2024-11-26
> **reply**
>
> Thanks for the responses. I keep my score due to four reasons:
>
> a) the related work, although has been refined, still shows imcompleteness. Two recently proposed surveys [1][2] might help the authors to futher grasp up-to-date works.
>
> b) the results in your response (W4-2), lack error bars. I can not tell the significance of the reulsts. Besides, the results for training GS-MODAC on 25j5m are missing. Further, if I understood it correctly, current results indicate that training on GS-MODAC on all sizes degrades the final performance? (e.g., 5.63 v.s. 5.70 on 10j5m, 2.09 v.s. 2.14 on 25j5m) I think this verifies the limited generalization of GS-MODAC.
>
> c) in W4-1, you claim that for new algorithm, only the final MLP requires to be changed. I agree with that, however, what shape of the MLP (number of layers, what kind of activation function) should be adopted when we adapt GS-MODAC to different algorithms with different complexities? Besides, although the network before the final MLP does not to be changed, it requires re-training, this is not efficient. If the authors claim universality of GS-MODAC across diverse algorithms, there are two key points should be integrated into current version (from my perspective): 1) making the MLP a fixed maximum output length, while interprete action for different algorithms. 2) training GS-MODAC on an associate distribution of algorithms and problem intances.
>
> [1] https://papers.ssrn.com/sol3/papers.cfm?abstract_id=4956956
> [2] https://arxiv.org/pdf/2411.00625

---

> ### Author Response · Authors · 2024-11-28
>
> Thank you for your response. Please find the revised paper attached as PDF.
>
> **Point A: Related work**
>
> Thank you for bringing these very recent works to our attention. We have carefully reviewed these papers and found valuable insights into challenges that require careful consideration and were addressed in our study. For instance, [1] discusses the difficulties of deep reinforcement learning (DRL) in managing high-dimensional state spaces and large-scale multitask learning problems. [2] highlights the limitations of many studies that overlook the parameter configurations of operators they learn to select, as well as the limitations in generalizability, where methods are often only trained and demonstrated on simple continuous optimization problems and standard benchmark functions. In our work, we focus on multi-objective combinatorial optimization. We build upon an N-dimensional graph as MDP, which avoids relying on high-dimensional state spaces while optimizing N objectives simultaneously. Additionally, we aim to learn how to control parameter configurations for solving combinatorial optimization. By configuring the state in the standardized graph-based representation, we can generalize trained models to different instance sizes, objectives, and complex problem variants. We have also shown the method's applicability in integrating it into different EC algorithms. We included the observed challenges in the related work section of our paper. We did not find any specific work in these two papers that is more relevant to our research than those we have already cited.
>
> **Point B: Significance**
>
> Please find the updated table below, including error bars, and the performance results of GS-MODAC trained on 25j5m and the overall-best baselines (SMAC3). With the results provided in the table above, we aimed to address your concerns regarding the scaling of the models trained on small problem instances (5j5m/10j5m, which are cheaper to train on) to solve large-sized instances.
>
> From the table, we observe that the model trained on all instance sizes performs less effectively than models trained directly on the size of the tested instances. However, it consistently outperforms all other baselines specifically trained or tuned on the testing problem configuration in terms of mean and maximum hypervolume (except for the maximum hypervolume on 5j5m, where irace scores slightly better). This supports the conclusion that the trained models can generalize effectively to handle instance sizes not encountered during training.
>
> | |**5j5m**|||**10j5m**|||**25j5m**|||
> |:-:|:-:|:-:|:-:|:-:|:-:|:-:|:-:|:-:|:-:|
> ||HV mean|HV max|HV std|HV mean|HV max|HV std|HV mean|HV max|HV std|
> |SMAC3|2.09×10^6|2.25×10^6|1.23×10^5|5.65×10^6|6.05×10^6|2.91×10^5|2.07×10^7|2.20×10^7|1.01×10^6|
> | GS-MODAC - 5j5m|2.10×10^6|2.25×10^6|1.16×10^5|5.51×10^6 |5.91×10^6|3.06×10^5 |2.03×10^7|2.16×10^7|1.07×10^4|
> | GS-MODAC - 10j5m|2.09×10^6 |2.25×10^6|1.24×10^5|**5.70×10^6**| **6.09×10^6**|2.99×10^5|2.13×10^7|2.26×10^7|1.06×10^6|
> |GS-MODAC - 25j5m|2.08×10^6|2.23×10^6|1.18×10^5|5.69×10^6|6.01×10^6|2.88×10^5|**2.14×10^7**|**2.27×10^7**|1.09×10^6|
> |GS-MODAC - all|**2.11×10^6**|2.23×10^6|1.17×10^5|5.63×10^6|6.05×10^6|3.18×10^5| 2.09×10^7|2.23×10^7|1.09×10^6|
> |GS-MODAC|2.10×10^6|2.25×10^6|1.16×10^5|**5.70×10^6**|**6.09×10^6**| 2.99×10^5|**2.14×10^7**|**2.27×10^7**|1.09×10^6|
>
> **Point C: Generalizability**
>
> It is important to clarify that our objective was not to create a universal model capable of managing multiple EC algorithms simultaneously. This is due to the significant differences in the search dynamics of various EC methods. For instance, Swarm Intelligence and Evolutionary Algorithms exhibit fundamentally different search behaviors, meaning that identical action selections can produce vastly different outcomes depending on whether they are applied with MOPSO or NSGA-II operators. To illustrate this, in response to point 1, we applied a trained model for MOPSO with a fixed output length to control NSGA-II in solving the CVRP. This resulted in a notable decrease in performance (see table below). In response to 2, we have trained GS-MODAC on both NSGAii and MOPSO simultaneously for different configurations of the CVRP problems. Also for this configuration, we observe a strong performance decline, although less severe than point 1.
>
> ||Bi-CVRP-100|||
> |-|-|-|-|
> |Method|mean|max|std|
> |GS-MODAC - NSGAii model|**1.35×10^2**|**1.48×10^2**|7.95|
> |GS-MODAC - MOPSO model (1)|1.17×10^2|1.31×10^2|9.46|
> |GS-MODAC - MOPSO & NSGAii model (2)|1.30×10^2|1.39×10^2|7.68|
>
> We acknowledge the need to retrain GS-MODAC when adapting it to a new algorithm may seem inefficient. However, this approach prioritizes preserving performance while ensuring versatility. The trained model can adapt effectively to problems configured with varying objectives, instance sizes, or more complex problem variants, scenarios that are highly relevant in real-world applications.

---

> > ### Comment · Reviewer_A1gs · 2024-11-30
> > **reply**
> >
> > I appreciate for the further explaination of the authors. However, the results presented above can not fully address my concerns (generalization). I will keep my score and have discussion with the 8-point reviewer and two 5-point reviewers after the rebuttal. I acknowledge this paper represents a promising direction, but the neural network design, training and the results are not compelling enough. refining these aspects could significantly strengthen this paper's quality.

---

> > > ### Author Response · Authors · 2024-11-30
> > >
> > > Thank you for your detailed feedback and for recognizing the promising direction of our work. We appreciate the time and effort you’ve dedicated to reviewing our submission.
> > >
> > >  **[Results]** We demonstrate the performance of the proposed GS-MODAC method across a variety of problem types, instance sizes, and number of objectives, by using different evolutionary algorithms. The results show that GS-MODAC consistently outperforms or matches competitive baselines in this wide range of scenarios. The method performs particularly well on problems with larger search spaces and more objectives, where the choice of parameter configurations significantly impacts search efficiency. Notably, GS-MODAC achieves superior performance on the Penta-FJSP-25j5m problem, outperforming the best baselines (irace and MADAC, tuned and trained directly on the problem configuration) by **8.5%** and **5.7%** in terms of mean and max hypervolume, and surpassing the vanilla NSGA-II by **10.6%** and **10.8%**, respectively.
> > >
> > >  - Given that our results clearly show a competitive edge over baselines, what additional evidence or refinements would you consider necessary to make our approach more compelling?
> > >
> > >  **[Generalizability]** To evaluate the generalizability of GS-MODAC, we analyzed the performance of trained models across varying configurations, including problem sizes, problem complexities, and objective configurations. The method demonstrates strong generalization capabilities by achieving competitive results even for configurations unseen during training. For instance, the 10j5m-trained model scales effectively to larger instances and more difficult problem variants (DAFJS and YFJS with SDSTs), while a model trained on all sizes performs robustly across configurations. Additionally, in response to your suggestion, we visualized the relationship between states and actions in Appendix G to provide an intuitive analysis of how the method adjusts its actions based on the graph's features. This demonstrates GS-MODAC’s flexibility and adaptability, which play a crucial role in its overall success and versatility.
> > >
> > >  - Could you elaborate on your remaining concerns about the generalization of our approach after the rebuttal? We are eager to address any specific points that may not have been fully clarified.
> > >
> > >  **[Network design and training]** We performed extensive experimentation on the network design of GS-MODAC, exploring several alternative configurations, as outlined in the ablation study in Appendix E. This included testing more advanced techniques such as attention mechanisms and transformers, but these did not improve performance over the selected design. Regarding training, it is important to emphasize that the computational overhead for both training and tuning is primarily driven by the underlying evaluation function of the generated solutions. Notably, the training of our method for the largest FJSP problem variant was approximately 35% more computationally expensive than the SMAC3 method. However, once the network is trained, it can be deployed directly for testing without further adjustments. Regarding inference, our method incurs only a negligible computational overhead: 2% for the smallest FJSP configuration, which decreases to just 0.001% for the largest configuration, as the evaluation of the underlying solutions dominates the overall runtime.
> > >
> > >  - Do you have any concrete suggestions or advice that could help us further strengthen these aspects in ways that align with your expectations?

---

> > > > ### Author Response · Authors · 2024-12-02
> > > >
> > > > Dear Reviewer A1gs,
> > > >
> > > > The deadline for discussion period is approaching. We hope we have adequately addressed your concerns. If you have any additional questions you'd like to discuss, please feel free to contact us. Thanks for your precious time and effort!
> > > >
> > > > Best regards,
> > > >
> > > > Authors of Submission 6550

---

> > > > > ### Comment · Reviewer_A1gs · 2024-12-02
> > > > >
> > > > > I thank the authors for their efforts and responses. I rise my score to 5.

---

> > > > > > ### Author Response · Authors · 2024-12-02
> > > > > >
> > > > > > Dear Reviewer,
> > > > > >
> > > > > > Thank you for your kind words and for raising your score to 5. We truly appreciate your time and feedback.
> > > > > >
> > > > > > May we ask if there are any specific concerns preventing you from accepting our work? Your guidance would be invaluable in helping us address them.
> > > > > >
> > > > > > Best regards,
> > > > > >
> > > > > > Authors of Submission 6550

---

### Official Review · Reviewer_RHLS · 2024-11-03

**Soundness:** 3
**Presentation:** 3
**Contribution:** 3
**Rating:** 8
**Confidence:** 4

**Summary:**

This paper describes a GNN based DRL approach to configurating multi-objective evolutionary algorithms for combinatorial optisation problems. Flexible job shop scheduling and capitalised Vehicle Routing problems are considered in the paper. The results show that the proposed method achieved better results than existing methods and has wider adaptability.

**Strengths:**

1. the paper considered graph representation in neural networks in DRL, which is good for complex combinatorial optimisation.

2. The motivations and contributions are very clear and relatively convincing.

3. The results and analyses are good.

**Weaknesses:**

1. A major weakness is that this paper only considered static/certain problems for FJSS and CVRP. It did not consider dynamic or uncertain problems, which are more difficult.

2. The shops and CVRP problems used are relatively small. For example, I have seen many FJSS problems use over 5000 jobs.

**Questions:**

Genetic programming and other OR methods are widely used to solve JSS and VRP problems. Why did not you consider comparisons with those commonly used methods in the paper?

---

> ### Author Response · Authors · 2024-11-23
>
> Thank you to the reviewer for their thoughtful review and helpful suggestions. We have provided our responses to the comments below and attached the updated manuscript as a PDF.
>
> **W1: Choice for static problems**
>
> Although we acknowledge that stochastic problems are very interesting and relevant, addressing these scenarios using search-based methods typically requires extensive simulations to account for variability, vastly increasing the computational resources required for this study. By sticking to the selected problems in this work, we were able to highlight its effectiveness in dynamically configuring algorithms based on the stage of the search, and generalizing to different problem sizes (Table 2), more complex problem variants (Table 3), and different objective functions (Table 4).
>
> **W2: Size of Shops and CVRP problems**
>
> We focused on smaller problem instances to assess the effectiveness of GS-MODAC. These instances enabled controlled experiments, demonstrating the method's applicability across diverse problem domains, sizes, and objectives. Scaling up to larger problem instances would significantly increase computation time, making search-based methods a less optimal solution.
>
> **Q1: Selection of benchmarking methods**
>
> In this work, we focus on the dynamic configuration of algorithms. As such, we have compared with traditional methods for algorithm configuration (vanilla (NSGAii), Bayesian optimization (SMAC3), and search (iRace), and with state of the art dynamic algorithm configuration method (MADAC).

---

> > ### Author Response · Authors · 2024-12-02
> >
> > Dear Reviewer RHLS,
> >
> > The deadline for discussion period is approaching. We hope we have adequately addressed your concerns. If you have any additional questions you'd like to discuss, please feel free to contact us. Thanks for your precious time and effort!
> >
> > Best regards,
> >
> > Authors of Submission 6550

---

> > > ### Author Response · Authors · 2024-12-03
> > >
> > > Dear Reviewer RHLS,
> > >
> > > The author-reviewer discussion deadline is approaching (in 8 hours). Please kindly let us know if we have addressed your concerns. We would appreciate it if you could give us any feedback. Thanks for your valuable time to review our rebuttal!
> > >
> > > Best regards,
> > >
> > > Authors of Submission 6550

---

### Official Review · Reviewer_GDac · 2024-11-03

**Soundness:** 2
**Presentation:** 2
**Contribution:** 2
**Rating:** 5
**Confidence:** 4

**Summary:**

In this work, the authors propose a DRL (Deep Reinforcement Learning)-based dynamic algorithm configuration method, GS-MODAC, to solve multi-objective combinatorial optimization (MOCO) problems. When modeling the parameter configuration problem as a MDP (Markov Decision Process), GS-MODAC employs graph neural networks (GNNs) to learn state representations, which distinguishes GS-MODAC from related studies. Additionally, a shared reward function is developed and it is applicable to different problem instances. The authors conduct a set of experiments on the variants of two MOCO problems, showing the effectiveness of the proposed algorithm and its generalizability.

**Strengths:**

1. The explanation of background and motivation is clear.
2. The writing is well-organized.
3. The authors conduct experiments from different perspectives to evaluate the performance of their algorithm.

**Weaknesses:**

1. Graph-based state representation is one of the main contributions in this work (as listed in the end of Section 1). In Section 3.1, the paragraph of states, the authors provide some explanations about how to construct the graph-based state representation and emphasize that this state configuration is effective. Further explanation is expected to clarify why this representation is more effective than alternatives.

2. How do states and rewards interact with the DRL agent? In Section 3, there is little information about the DRL agent in GS-MODAC. The authors can add a brief overview of how the DRL agent uses the state and reward information to learn and make decisions. Alternatively, the authors can add a brief introduction to DRL in Section 2.

3. In Section 3.2, many operations are described without details and explanations. For example, in line 266, what is the reason for introducing a global mean pooling operation to average the node embeddings? Similarly, the reason for including an additional feature vector and how this feature vector is incorporated into the state representation should be explained. The authors should provide brief justifications for each component design, and explain how each component contributes to the overall performance of GS-MODAC.

4. I cannot observe the results from Table2, as concluded by the authors on page 8, RQ2:
    4.1. The authors claim that “the model trained on smaller instances and deployed on larger instances experienced a slight decline in performance but still managed to find better solutions compared to all the benchmarks provided.” What do the authors mean by “all the benchmarks”? Are they referring to the comparison algorithms? If so, as reported in Table 1, GS-MODAC that trained on Bi-CVRP - 100 customers performs worse than MADAC that trained on Bi-CVRP - 500 customers.
    4.2. In addition, the authors claim that “the models trained on larger instances and deployed on smaller instances could improve results further”, but models trained on Bi-CVRP - 200 customers and deployed on Bi-CVRP - 100 customers found similar instead of better solutions than models directly trained on these smaller-sized instances. Moreover, models trained on Bi-CVRP - 500 customers and deployed on Bi-CVRP - 100 customers found worse solutions than models directly trained Bi-CVRP - 100 customers.
The authors should clarify their claims with more specific evidence from Table 2.

5. In Table 3, only the baseline NSGA-II is compared, which is not very persuasive. To demonstrate effectiveness, I suggest adding other AC/DAC algorithms and the GS-MODAC directly trained on these two complex problems as comparisons.

6. In Table 6, the experiments are conducted on CVRP problems of sizes 20, 50, and 100. Why not keep the settings consistent with the previous NSGA-II experiments (100, 200, and 500)? Is there any specific reason for this choice? Furthermore, I suggest including other AC/DAC comparison algorithms in this set of experiments.

7. In Table 7, the most related work MADAC should be compared.

8. The names of problem instances are not consistent. some CVRP problems are denoted as Bi-CVRP-100 customers in Table 1, but are denoted as Bi-CVRP-100 nodes or Bi-CVRP-100 n in Table2.

**Questions:**

1. GS-MODAC requires an additional run of EAs with a high budget (e.g. doubled) to obtain $HV_{ideal}$. Is this realistic in real-world applications? How does a double-budget NSGA-II perform when compared with the proposed GS-MODAC ?

2. In Figure 2, why wasn't SMAC3 included in the comparison? According to the results reported in Table 1, SMAC3 outperforms NSGA-II on Tri-FJSP 10j5m.

3. In Table 9, why is the $HV mean$ not used as a performance indicator, just as reported in the previous tables?

---

> ### Author Response · Authors · 2024-11-23
>
> We appreciate the insightful feedback. Below are our detailed responses, and the revised paper is attached.
>
> **W1: Graph representation benefits**
>
> We used a graph representation to avoid the cumbersome and suboptimal process of manual state space design. We have added additional explanations of its benefits in the introduction and provided more details about its construction in the caption of Figure 1 (marked in red).
>
> **W2: DRL agent interaction**
>
> Following the reviewer's suggestion, we have extended subsection 3.2 to briefly reflect on how the agent interacts with the MDP.
>
> **W3: Components justification:**
>
> Pooling aggregates the node embeddings into a fixed-size graph representation, summarizing the population’s solutions to predict algorithm parameters. To investigate the mean pooling choice, we tested max, sum, and attention pooling methods on Tri-FJSP-10j5m and found them generally inferior to the mean pooling used in our work:
>
> ||**Tri-FJSP**||
> :-:|:-:|:-:
> ||HV mean|HV max
> max pooling|5.67×10^6|6.05×10^6
> sum pooling|5.61×10^6|6.04×10^6
> attention pooling|5.68×10^6|**6.09×10^6**
> GS-MODAC|**5.70×10^6**|**6.09×10^6**
>
> The feature vector, indicating the remaining search budget, is provided to the agent after the pooling layer. Its impact is shown in Table 7, which also highlights the benefit of multiple GCN layers. Table 8 compares GCN's contribution to other popular GNNs. In an additional experiment, we tested the HV rewarding component with an Inverted Generational Distance (IGD) metric. Results in Appendix E demonstrate GS-MODAC's effectiveness for IGD.
>
> **W4-1: Results observation:**
>
> We are indeed referring to the comparison algorithms. Although GS-MODAC, trained on Bi-CVRP-100, performs slightly worse than MADAC (trained on Bi-CVRP-500) in mean HV, it outperforms MADAC in max HV. We’ve updated our statement (in red in Section RQ2) to clarify that the generalized models, despite training on different problem sizes, perform similarly to MADAC while surpassing the other baselines.
>
> **W4-2:  Results observation:**
>
> The reviewer’s observation is correct. We have revised the statement in RQ2 to emphasize that models trained on diverse instance sizes can learn a robust, well-performing policy. We ran additional experiments and added performance results for a model trained on all Bi-CVRP sizes to Table 2. Additionally, we provide below new results for a model trained on all Tri-FJSP sizes, outperforming the model trained on the smallest instances for 5j5m, and the best non-learning baseline (irace) on the larger instances.
>
> ||**5j5m**||**10j5m**||**25j5m**||
> :-:|:-:|:-:|:-:|:-:|:-:|:-:|
> ||HV mean|HV max|HV mean|HV max|HV mean|HV max
> irace|**2.11×10^6**|**2.26×10^6**|5.47×10^6|5.82×10^6|2.07×10^7|2.20×10^7
> GS-MODAC|2.10×10^6|2.25×10^6|**5.70×10^6**|**6.09×10^6**|**2.14×10^7**|**2.27×10^7**
> GS-MODAC - all|**2.11×10^6**|**2.26×10^6**|5.63×10^6|6.05×10^6|2.09×10^7|2.23×10^7
>
> **W5: Additional Baselines**
>
> We added new baseline results in Table 3 (in red), including GS-MODAC trained on the complex variants. GS-MODAC trained on the complex variant demonstrates superior performance (except for mean HV in the Penta-problems). The results also show that the method generalizes well to more challenging variants. Notably, GS-MODAC trained on 10j5m outperforms all other baselines tuned for the DAFJS and YFJS problems.
>
> **W6: Problem size CVRP**
>
> We conducted the experiment using the PMOCO setup, including their tuned models, configurations, data generation, and problem sizes (20, 50, and 100). Following another reviewer's suggestion, we did additional experiments and updated Appendix F with results for NHDE-P (Chen et al., 2024). NHDE-P outperforms PMOCO in generalizability but underperforms on the smallest instance size and smaller different distributed instances.
>
> **W7: MADAC inclusion**
>
> Thanks for the suggestion; we’ve added MADAC to the ablation study in Table 7.
>
> **W8: Naming consistency:**
>
> We updated the names to use only the 'Bi-CVRP-100' notation.
>
> **Q1: High budget run**
>
> Our reward function requires only an approximation of the ideal point during training. Lower budgets can still provide a good estimate, and even with an underestimated ideal point, the function remains effective, rewarding convergence beyond 100%. The table below shows that NSGA-II with a doubled budget does not outperform GS-MODAC.
> ||**25j5m**|||
> |:-:|:-:|:-:|:-:|
> ||HV mean|HV max|HV std
> NSGAii (double)|2.10×10^7|2.23×10^7|1.12×10^6
> GS-MODAC|**2.14×10^7**|**2.27×10^7**|1.09×10^6
>
> **Q2: SMAC3 comparison**
>
> We selected NSGAii as the baseline for its foundational configuration. However, since SMAC3 performs better for this problem, we conducted further experiments and included it in Figure 2. The figure shows GS-MODAC outperforming both baselines.
>
> **Q3: HV performance metric**
>
> We conducted the experiment using the PMOCO setup, including their tuned models, configurations, data generation, and problem sizes (20, 50, and 100).

---

> > ### Comment · Reviewer_GDac · 2024-11-26
> >
> > Thank you for the thorough response.
> >
> > There are some follow-up questions:
> > 1.  The working mechanism of DRL Agent remains unclear. After inputting states, what happens within the DRL Agent? The authors should provide additional background knowledge about the DRL Agent. This would also help readers understand the advantages of the proposed graph-based state. The revised introduction and the captions in Fig. 1 explain the disadvantages of alternative states and the workflow of GS-MODAC, respectively. However, these revisions do not clarify why the graph-based state is effective and why it has the advantages claimed by the authors.
> >
> > 2. The experiment reported in Table 4 shows that the performance of GS-MODAC is similar to NSGAii. To further demonstrate the performance of GS-MODAC, more comparisons with algorithms such as irace and SMAC3 should be included.
> >
> > 3. The names of competitors should be consistent. Please address all inconsistencies, such as 'smac3' and 'SMAC3' in Table 3.
> >
> > 4. The advantage of GS-MODAC is not significant in the results reported in Tables 9 and 10. Could the authors compare them on some FJSP problems to better demonstrate the effectiveness of GS-MODAC?
> >
> > 5. For Tables 9 and 10, results such as 'HV max' and 'HV std' should be reported, as was done in other result Tables.
> >
> > 6. For all experimental results reported in Tables, statistical tests are required to demonstrate that the advantages of GS-MODAC are significant.

---

> ### Author Response · Authors · 2024-11-28
>
> Thank you for your valuable feedback, which has helped improve our work. Below are our detailed responses to the follow-up questions, and a revised version is included in the attached PDF.
>
> **Q1: Agent mechanism**
>
> After receiving the state input, the DRL agent processes the graph-based state using two GCN layers. These layers extract node embeddings, capturing both local and global structural information within the graph. A global mean pooling operation then aggregates these node embeddings into a single graph-level embedding, creating a compact summary of the graph's structure. This graph embedding is concatenated with a feature vector containing search budget information and is then passed through a linear layer to predict the mean values of the action distributions. We have described this working in lines 272-284, and the advantages of the agent configurations are demonstrated through the ablation study in Appendix E.
>
> Graphs offer a powerful means of representing structured and informative embeddings, with the flexibility to scale to different sizes. Graph Neural Networks (GNNs), such as GCNs, have been widely applied across diverse graph-related tasks due to their ability to effectively model graph structures and extract meaningful representations [1]. In our case, we leverage GNNs to extract the graph state, enabling the DRL agent to make more informed and effective decisions based on the state of the iterative search. We modified section 3 to provide readers with essential background knowledge on GNNs and outline their benefits for our graph-based learning. Additionally, following the suggestion, we have incorporated additional background knowledge of DRL in related work in ‘algorithm configuration’.
>
> [1] https://www.sciencedirect.com/science/article/pii/S2666651021000012
>
> **Q2: Additional Baselines**
>
> We have included the additional baselines in Table 4. We observe here that the observation still holds; the trained models can be transferred to other configurations of the problem, finding solutions of similar or better quality than the configured baselines, with a similar performance gap as observed for two objective problem variants displayed in Table 1.
>
> **Q3: Name consistency**
>
> Thanks for this observation, we updated the names to use only the 'SMAC3' notation.
>
> **Q4: Comparison on FJSP**
>
> In the context of multi-objective optimization, we are unaware of any end-to-end learning approaches specifically designed for multi-objective FJSP. We compared our method against end-to-end approaches such as PMOCO and NHDE-P, which are designed specifically for multi-objective routing problems. Although our method did not outperform PMOCO and NHDE-P on all instances, we want to point out that POMCO and NHDE-P can only be used to solve specific routing problems, as they rely on specialized network architectures tailored for  Traveling Salesman Problem (TSP) and Capacitated Vehicle Routing Problem (CVRP).  In contrast, our method effectively guides multi-objective evolutionary algorithms, enabling them to address diverse problems across different domains and with varying objectives. This versatility allows our approach to be seamlessly applied to two distinct problem types (scheduling and routing), a capability that has yet to be demonstrated by other Pareto front approximation/end-to-end solution methods.
>
> **Q5:  HV performance metric**
>
> We formulated the results of this experiment using the PMOCO setup, including their objective, tuned models, configurations, data generation, and problem sizes (20, 50, and 100). Following another reviewer's suggestion, we did additional experiments and updated Appendix F with results for NHDE-P (Chen et al., 2024), which was also performed according to this setup.
>
> **Q6: Statistical tests**
>
> We have performed statistical tests using the Wilcoxon rank-sum test ($p < 0.05$), as explained in Section 4 in “testing” and we demonstrate significance in the main results table (Table 1). Here, we underlined significantly better solution values. In the additional tables, we included alternative baselines for GS-MODAC, such as models trained on different problem sizes or variants. In these tables, we observed for various configurations that proposed GS-MODAC models performed significantly better than other baseline methods. However, as there is no significant difference among the GS-MODAC variants themselves, we could not identify a GS-MODAC configuration that significantly outperformed all the included solution results. Therefore, no underlining is performed for these scenarios.

---

> > ### Author Response · Authors · 2024-12-02
> >
> > Dear Reviewer GDac,
> >
> > The deadline for discussion period is approaching. We hope we have adequately addressed your concerns. If you have any additional questions you'd like to discuss, please feel free to contact us. Thanks for your precious time and effort!
> >
> > Best regards,
> >
> > Authors of Submission 6550

---

> > > ### Author Response · Authors · 2024-12-03
> > >
> > > Dear Reviewer GDac,
> > >
> > > The author-reviewer discussion deadline is approaching (in 8 hours). Please kindly let us know if we have addressed your concerns. We would appreciate it if you could give us any feedback. Thanks for your valuable time to review our rebuttal!
> > >
> > > Best regards,
> > >
> > > Authors of Submission 6550

---

> > > > ### Comment · Reviewer_GDac · 2024-12-03
> > > >
> > > > Thanks for your response.
> > > >
> > > > Two quick questions:
> > > > 1. Are statistical tests performed for the results in all tables or only in Table 1? I found no underlines in the tables except for Table 1.
> > > > 2. One shortcoming of this work is the limited number of baselines (only 3–4). I am curious about the exclusion of MADAC from many experiments and tables, particularly in Tables 3 and 4.
> > > >
> > > > I will discuss with other reviewers in the reviewer's discussion phase.

---

> ### Author Response · Authors · 2024-12-03
>
> Thank you for this response. Below, we answer your questions:
>
> **[Statistical tests]** We conducted statistical tests for the results presented in the remaining tables. However, many of these experiments included alternative GS-MODAC-based baselines, which were trained on different problem sizes or variants. Since the performance of various GS-MODAC-based models was similar, there were no significant differences among these variants. As a result, no single GS-MODAC configuration consistently outperformed all other solutions. We have identified testing scenarios where all trained models significantly outperformed the included baselines (e.g., in the Tri- and Panta DAFJS and YFJS problem variants).
>
> **[MADAC baselines]** As stated in lines 57-60, we argue that MADAC, being designed for continuous optimization, may not work well on large-size, complex COPs with many objectives, due to less smooth solution spaces and a wide range of objective values of COPs. This limitation was evident in its substantial performance shortcomings across various scheduling problem variants in Table 1. Consequently, we excluded MADAC from the additional scheduling experiments presented in Tables 3 and 4. Notably, the other baseline methods that were included in these tables consistently match or surpass MADAC's performance on the scheduling problems presented in Table 1 (apart from max HV for Penta-FJSP-25j5m).

---

### Official Review · Reviewer_Y87f · 2024-11-04

**Soundness:** 2
**Presentation:** 2
**Contribution:** 2
**Rating:** 5
**Confidence:** 5

**Summary:**

This paper proposes a DRL method based on GNN, named GS-MODAC. The method models the algorithm configuration problem as a Markov decision process and leverages GNN to learn embeddings of solutions in the objective space, thereby enhancing state representation. However, there are some issues that need to be improved.

**Strengths:**

1. The idea is clear.
2. The authors provide experiments.

**Weaknesses:**

1. The current status of research lacks in-depth analysis, and the problems and key challenges that this research aims to address have not been accurately summarized.
2. In each iteration, the state is represented by embedding information from the current population solutions using a GNN, effectively capturing the structure and diversity of the solutions. However, GCN is transductive GNN, it has limited transferability to new graph structures. In other words, the GNN requires retraining on the graph structure with each iteration of the algorithm, which leads to significant computational costs. The authors should take this issue into account.
3.  As illustrated in Figure 1, each DRL iteration is essentially based on the transition between the current population state and the next iteration's population state. Specifically, the DRL strategy generates new parameter configurations based on the current state (i.e., the performance of the current population), which guide the optimization algorithm's search in the next iteration. This approach requires updating the DRL parameters at each iteration, which similarly incurs substantial computational costs.
4.  Considering comments 2 and 3, the computational complexity of this method is high. However, this paper does not include an analysis of the complexity.
5. The state configuration embedding method based on GCN is a key aspect of this research. However, Figure 1 does not effectively highlight the central role of GCN.
6. The figures provided in this paper, including the framework and experimental result, are not clear. Please provide high-resolution versions.
7. The conclusion lacks an in-depth analysis of the limitations of the this research and potential directions for future improvements. It is suggested that the authors include relevant content on these aspects.

**Questions:**

see Weaknesses

---

> ### Author Response · Authors · 2024-11-23
>
> We appreciate the reviewer's effort and insightful feedback. Here are our detailed responses. We provide a revised version of our work in the attached PDF file.
>
> **W1: lack in-depth analysis**
>
> **Aim** As described in our abstract, this work develops a DRL-based method for dynamic algorithm configuration (DAC) in multi-objective combinatorial optimization (MOCO). Unline static algorithm configurations, which assume fixed optimal parameters, DAC adapts during optimization. Despite its effectiveness, its research on MOCO remains underexplored. This work innovatively applies DRL with graph neural networks to automatically learn DAC for MOCO.
>
> **Challenges** Applying DRL to learn DAC for MOCO poses significant challenges. Configurations at each step must be guided by states and rewards that capture effects across multiple objectives. Existing RL-based DAC approaches have limitations: they often rely on manually selected convergence and landscape features, which are tedious and suboptimal. Moreover, they primarily focus on operator selection while neglecting the operator's parameters, and are limited to simple continuous optimization problems, overlooking the greater complexity of MOCO's discrete and combinatorial solution spaces. We discussed this in lines 51-65.
>
> **Solutions** We propose a graph-based representation framework that eliminates manual feature design to address these challenges. Inspired by metrics like elite solutions, solution spacing, gaps, and hypervolume, our method dynamically learns advanced features to capture the state across objective planes. With normalization in graph representation and reward functions, our approach achieves superior performance in configuring MOCO algorithms, generalizing effectively across problem sizes (Table 2), complex variants (Table 3), and objectives (Table 4). We discussed this in lines 60-65
>
> **W2: Embedding information in a GNN**
>
> Unlike the reviewer suggested, we do not retrain our graph structure during each iteration. Our approach focuses on graph classification and uses GCNs in an inductive manner. Specifically, we separate the train and test data, training the GCN only on graphs from the algorithm’s search state for training instances. Empirically, we have shown that our inductive approach enables the GCN to achieve strong generalization performance, as demonstrated by our results (Tables 2, 3 & 4). Intuitively, the GCN learns effective graph embeddings from the (normalized) structure of Pareto solutions that differentiate states in the algorithm process.
> Our results show that the proposed method is computationally efficient (as detailed in responses to W3 and W4 below), and delivers promising outcomes in the targeted optimization tasks. We conducted an ablation study to evaluate the choice of GCN (Table 8), which consistently delivered the best overall performance with comparable runtime, confirming its effectiveness for our framework.
>
> **W3: Computational costs of updating DRL parameters**
>
> Unlike the reviewer mentioned, our approach does not require updating the DRL parameters at each iteration. After training, the inference of the DRL policy only requires the state configuration at each step and the computation of the policy network to generate new parameter configurations for the optimization algorithm. Compared to the static-configured algorithm, the added computational costs of GS-MODAC are negligible, especially for larger problem instances where the additional computation of GS-MODAC is more slight than the runtime of the optimization algorithm itself. Further details on the computational costs of our approach and its components can be found in our responses to W4.
>
> **W4: Computational Complexity**
>
> We profiled GS-MODAC to assess its computational complexity, focusing on graph state configuration and policy network inference. Results show the actor's inference time is 0.13 seconds, and state extraction takes 0.2 seconds, together accounting for 2.0% of the total time for the smallest problem instances. For larger problems, this proportion decreases significantly as solution evaluations dominate computation. Despite a slight overhead, its substantial performance gains justify GS-MODAC’s minimal additional cost.
>
> ||Bi-FJSP-5j5m|Penta-FJSP-5j5m|Bi-FJSP-25j5m|Penta-FJSP-25j5m|
> |:-:|:-:|:-:|:-:|:-:|
> |total time|15.09s|15.46s|305s|302s|
> |total state configuration time|0.18s|0.21s|0.23s|0.22s|
> |total policy inference time|0.12s|0.13s|0.14s|0.13s|
>
> **W5&6: Role of GCN and Resolutions**
>
> Figure 1 has been updated with detailed computations within the agent, and the resolutions of all figures have been improved.
>
> **W7: Analysis of limitations**
>
> A limitation of our approach is the lack of usage of a specialized RL algorithm for contextual MDPs. While our ablation study showed the advantage of GCN, it can be further advanced to enhance the learning of Pareto front representations. We have updated our conclusion section with this analysis.

---

> ### Comment · Reviewer_Y87f · 2024-11-26
>
> After reading other reviews and the rebuttal, I hold my score due to following reasons:
> 1. Please provide an in-depth analysis of your method, your responses are not convincing. For example, you should provide a theoretical analysis to demonstrate the effectiveness of your method.
> 2. Exploring the reasons behind the success of these techniques and providing intuitive explanations would contribute to the overall scientific contribution of the work.
> 3. Why not provide computational complexity before rebuttal?
> 4. The format of references violates the code of ICLR.
> 5. Please provide more peer competitors to enhance the generalizability of your method.
> 6. You should provide the training costs, and memory costs.  DRL is very slow compared to other superior methods.

---

> ### Author Response · Authors · 2024-11-28
>
> Thank you for these insightfull points. Here are our detailed responses. We provide a revised version of our work in the attached PDF file.
>
> **Q1: In-depth analysis**
>
> Theoretical analysis for methods that use neural networks (NNs) to learn heuristics (DRL policies) is very rare in existing literature. This is because NNs learn complex patterns that are difficult to explain or model mathematically. The adaptive nature of NNs makes it hard, often impossible, to provide a detailed theoretical justification for their performance.
>
> Intuitively, our method works well because the DRL policy learns to leverage the graph-based representation to capture the current search state of the iterate process, sample actions to interact with it, and achieve better convergences of the different objectives. The reward scheme is designed to encourage Pareto optimal solutions in a problem-agnostic way, with increasing rewards as the search converges toward an ideal point. By incorporating normalization in both the graph representation and the reward function, we ensure the method generalizes well and performs efficiently across various scenarios. For more in-depth insights, we refer to our response to Q2, where we offer a further intuitive explanation of the method’s effectiveness, supported by additional analysis in Appendix G.
>
> **Q2: Intuitive explanations behind the success of GS-MODAC**
>
> The success of our method can be attributed to its ability to adapt actions based on the underlying graph states, even without direct knowledge of the encoded solutions, instance information, or the objective to be optimized. To illustrate this, we visualized the relationship between states and actions in Appendix G, using a model trained on objectives A/B to solve both A/B and C/D objectives configurations. The figure shows that while the graph patterns for A/B and C/D are visually similar, their convergence behaviors differ. A/B converges more quickly toward balanced solutions, while C/D tends to produce solutions with more competing objectives, resulting in wider Pareto fronts. These differences lead to varying action strategies: for A/B are high mutation and low crossover rates selected, while for C/D it selects higher crossover rates.
>
> The different graph states offer an intuitive explanation for this. The A/B solutions encoded in the graph, which are more similar in objective values, benefit from more local exploration. In contrast, C/D's solutions, which involve competing objectives, require a broader exploration of the solution space. This ability of the method to adjust actions based on the graph's features demonstrates its flexibility and adaptability, contributing significantly to the overall success of the approach.
>
> **Q3: Reporting on computational complexity**
>
> Computational complexity is often not included in the literature for neural network-based methods like ours, as the emphasis generally tends to be on empirical performance rather than computational complexity. In our case, we chose not to report inference times initially because the algorithm adds only a negligible computational overhead: 2% for the smallest FJSP configuration and dropping to only 0.001% for the largest configuration, as the evaluation of the underlying solutions dominates the runtime. Following the reviewer's suggestion, we have included our computational complexity analysis in Appendix I.
>
> **Q4: References format**
>
> Thanks for this observation, we have updated the violating reference where we used parenthesis using \citet{} to meet the ICLR template requirements.
>
> **Q5: Peer competitors comparison**
>
> We have integrated our framework with a Swarm Intelligence-based algorithm, MOPSO, with implementation details provided in Appendix B. The results presented in Table 5 (Appendix C) show that GS-MODAC enhances MOPSO's performance, demonstrating its effectiveness across different algorithms.
>
> **Q6: Training and memory costs**
>
> Details about the training are provided in Section 4, under Baselines and Training. The training was conducted on an Intel(R) Xeon(R) CPU E5-1650 v4 processor with 16 GB of RAM, using five parallel environments. While DRL requires a longer training period than the tuning times of the baselines, the trained network can be directly utilized during testing without needing additional training or fine-tuning of the policy during inference. This makes it highly efficient in terms of testing and deployment.

---

> > ### Author Response · Authors · 2024-12-02
> >
> > Dear Reviewer Y87f,
> >
> > The deadline for discussion period is approaching. We hope we have adequately addressed your concerns. If you have any additional questions you'd like to discuss, please feel free to contact us. Thanks for your precious time and effort!
> >
> > Best regards,
> >
> > Authors of Submission 6550

---

> > > ### Author Response · Authors · 2024-12-03
> > >
> > > Dear Reviewer Y87f,
> > >
> > > The author-reviewer discussion deadline is approaching (in 8 hours). Please kindly let us know if we have addressed your concerns. We would appreciate it if you could give us any feedback. Thanks for your valuable time to review our rebuttal!
> > >
> > > Best regards,
> > >
> > > Authors of Submission 6550

---

### Author Response · Authors · 2024-11-25

We appreciate the reviewers for providing valuable feedback. In response to the feedback, we have addressed the comments point by point and made significant additions to the paper, which can be found in the revised paper attached. For your convenience, we list the main revisions below:

- **[Interpretability Analysis]:** We conducted an in-depth analysis in Appendix H to address interpretability concerns. We captured the obtained graph states and analyzed the relationship between these patterns and the action distributions controlled by GS-MODAC. This analysis provides insights into how GS-MODAC selects actions and highlights its ability to adapt action distributions to optimize objectives not encountered during training.

- **[Comparative Analysis of Graph-Based Methods]:** We conducted an in-depth analysis to compare GS-MODAC's GNN-based feature extraction with recent approaches Chen et al. (2023). This comparison, included in Appendix F, highlights the versatility of GS-MODAC’s graph-based representation and shows the performance advantages of GS-MODAC's method for generalizing to different distributed instances in the other sizes.

- **[Computational Complexity Profiling]:** We conducted a detailed profiling of GS-MODAC, demonstrating its efficiency with minimal computational overhead. For larger problems, this overhead becomes even less significant as solution evaluations dominate computation. Overall, GS-MODAC’s minimal added cost is well justified by its substantial performance gains.

- **[Component Validation]:** We performed additional ablations to validate the components of GS-MODAC, emphasizing the benefits of the pooling operation and its rewarding component. This builds on prior results in Appendix E, highlighting the advantages of the additional feature vector, multiple GCN layers, and the selected graph neural network type.

- **[Generalized Policy Performance]:** We demonstrate the ability of GS-MODAC to train a single robust policy across diverse instance sizes, highlighting the generalization capability of the policy across different problem sizes.

We believe the updates strengthen our work and better align it with the expectations of the community. As the deadline for author-reviewer discussion approaches, we kindly ask the Area Chair and reviewers to evaluate our work based on our point-by-point responses with the revised paper, and let us know if we have addressed the concerns.

If you have any additional questions you'd like to discuss, please feel free to contact us. Thanks for your precious time and effort!

---

### Meta-Review · Area_Chair_kLiT · 2024-12-19

**Metareview:**

The paper proposes a method for controlling the intermediate behaviors (e.g. crossover and mutation coefficients) of an evolutionary algorithm (e.g. NSGA-II) for multi-objective optimization. Given a standard evolutionary algorithm, the core idea is to construct a RL policy which:
  * Takes as input current trials (which construct a Pareto frontier), using a graph neural network (GNN)
  * Outputs intermediate coefficients which control the behavior of said evolutionary algorithm.

The RL policy is pretrained over various combinatorial problem instances (e.g. Flexible Job Shop Scheduling, Capacitated Vehicle Routing), and then tested on new problem instances.

Experimental results are standard, i.e. showing improvements over the base NSGA-II algorithm and improvements over other algorithms (irace, SMAC3) too.

## Weaknesses
The core idea seems incremental and lukewarm. It's not clear whether there's much to gain from mainly tuning the intermediate coefficients of evolutionary algorithms, especially with additional complexity (e.g. GNNs, RL, pretraining) and whether this leads to any deep fundamental insights.

Despite the authors' rebuttals, the need for heavy machinery such as GNNs also isn't very well-motivated - it's not obvious at all why such an architecture would be needed to process the multi-objective values, especially as they don't naturally form a graph. If the main intention is to simply ensure that the learnable parameters of the GNN are independent of the multiobjective space size, standard self-attention would have been just as fine.

**Additional Comments On Reviewer Discussion:**

Reviewers gave a post-rebuttal score of (5,5,5,8), making this a lukewarm paper.

The main issues raised align with my overall assessment, namely that (1) Novelty and contributions aren't significant enough and (2) Unclear why GNNs are required. A few additional points raised by reviewers and my takes:
  * Reviewer A1gs questions whether the RL policy needs to be trained for different optimizers and problems.
    * Since the RL action dimension depends on the number of optimizer coefficients, retraining will be required here, although since the authors use a GNN, it might not require retraining over new problem instances with different objectives, I believe.
  * Reviewer GDac raises experimental issues (e.g. baseline comparisons are lacking), although the authors seem to have provided additional experiments during rebuttal.

Reviewer RHLS gave an 8, but their review was very short and did not participate in additional rebuttal discussions, lowering their consideration as well.

Since most of the reviewers vote to reject, we will do so.

---

### Decision · Program_Chairs · 2025-01-22

Reject